# SENSITIVITY-CONSTRAINED FOURIER NEURAL OPERATORS FOR FORWARD AND INVERSE PROBLEMS IN PARAMETRIC DIFFERENTIAL EQUATIONS

**Abdolmehdi Behroozi & Chaopeng Shen** [*]
Department of Civil and Environmental Engineering
Penn State University
University Park, PA 16802-1408, USA
`{amb10399, cxs1024}@psu.edu`

**Daniel Kifer**
School of Electrical Engineering and Computer Science
Penn State University
University Park, PA 16802-1408, USA
`duk17@psu.edu`

## ABSTRACT

Parametric differential equations of the form $\frac{\partial u}{\partial t} = f(u, x, t, p)$ are fundamental in science and engineering. While deep learning frameworks like the Fourier Neural Operator (FNO) efficiently approximate differential equation solutions, they struggle with inverse problems, sensitivity calculations $\frac{\partial u}{\partial p}$, and concept drift. We address these challenges by introducing a novel sensitivity loss regularizer, demonstrated through Sensitivity-Constrained Fourier Neural Operators (SC-FNO). Our approach maintains high accuracy for solution paths and outperforms both standard FNO and FNO with Physics-Informed Neural Network regularization. SC-FNO exhibits superior performance in parameter inversion tasks, accommodates more complex parameter spaces (tested with up to 82 parameters), reduces training data requirements, and decreases training time while maintaining accuracy. These improvements apply across various differential equations and neural operators, enhancing their reliability without significant computational overhead (30%–130% extra training time per epoch). Models and selected experiment code are available at: `https://github.com/AMBehroozi/SC_Neural_Operators`.

## 1 INTRODUCTION

Ordinary and Partial Differential Equations (ODEs and PDEs), which contain physically meaningful parameters $\mathbf{p}$, form the foundation of most modern engineering systems across diverse fields such as fluid mechanics, medicine design, molecular dynamics, relativistic physics, climate, and environmental sciences (Palais & Palais, 2009). These physical parameters $\mathbf{p}$, whether describing fixed or evolving system properties (Feng et al., 2022), physiological constants (Reichert & Omlin, 1997), environmental characteristics (Tartakovsky et al., 2020), or serving as simplifications for subgrid-scale processes (Yuval & O'Gorman, 2020; Watt-Meyer et al., 2024), significantly influence the behavior of the equations. *Neural Operators* are neural networks whose inputs are initial conditions and physical parameters, and whose output is a function $\mathbf{u}$, also known as a solution path, that approximates the solution to the ODE/PDE. When the values of $\mathbf{p}$ are known, neural operators can be used for *simulations* (predicting system state $\mathbf{u}(t)$ at time $t$). However, in many cases, there is uncertainty about the values of $\mathbf{p}$, so they must be estimated from observations of $\mathbf{u}$. This is known as (parameter) *inversion*. The estimated $\mathbf{p}$ can then be used to simulate $\mathbf{u}$ beyond existing observations, or they may themselves be quantities of interest to scientists. Other potential uses of neural operators include *sensitivity* analysis (i.e., estimating $\partial \mathbf{u}/\partial \mathbf{p}$) and applying them in situations not covered by training data. Even state-of-the-art neural operators struggle with these applications and this paper presents a method to address this problem.

**Fourier Neural Operators (FNOs).** Introduced in 2021, FNOs (Li et al., 2021; 2024) leverage the linear nature of differential operators in Fourier space, delivering orders of magnitude of computational efficiencies compared to traditional numerical solvers. They are currently the most popular type of neural operator. FNOs are known for their meshless—or discretization-invariant—characteristics,

---
[*]Corresponding author

and some versions eliminate the need for time-stepping, allowing them to output solutions at any spatiotemporal resolution. FNOs achieve these improvements over prior surrogate models (Audet et al., 2000; Alizadeh et al., 2020) by utilizing representations in the *frequency domain* to efficiently learn time trajectories of the evolution of initial conditions. Learning the trajectories in *time domain* would necessitate large amounts of weights and extensive training data.

While this method offers significant efficiency, prediction accuracy may decline for predictions far into the future (Grady et al., 2023), prompting the use of additional constraints in some models (Jiang et al., 2023; Bonev et al., 2023). FNO is so far mostly used in forward simulations and research; the critical issues of parameter inversion have received much less attention by neural operator research and estimating the parameter sensitivity $\partial \mathbf{u}/\partial \mathbf{p}$ has not been studied for neural operators (to the best of our knowledge). Li et al. (2021) evaluate the use of FNO to run the differential equation backward in time to recover initial conditions; although they name it *Bayesian inversion*, they did not recover physical parameters. In follow-up work, (Li et al., 2023) also observed the problems of using FNO for parameter inversion and proposed to add physics-informed neural network (PINN) regularization to training. While it improves over vanilla FNO, we show that our approach can outperform FNO+PINN by up to 30%. Furthermore, FNO+PINN still produces inaccurate sensitivity estimates. An alternative to neural operators is learning direct inverse mapping (Vadeboncoeur et al., 2023; Li et al., 2023), but then one still needs separate techniques for estimating solution paths and sensitivities. In contrast, SC-FNO addresses all of these issues in one (convenient) framework.

**Sensitivity awareness** allows a model to adapt its predictions to different scenarios and respond to input and parameter changes, thus reducing overfitting and improving robustness. In practical applications, the ability to predict these Jacobians of the outputs with respect to physical parameters $\mathbf{p}$ can also have significant value. In engineering design and control systems, knowing the sensitivity of performance outputs relative to design parameters helps in optimizing design and improving performance. Similarly, in financial modeling or resource management, sensitivity information allows for better risk assessment and resource allocation decisions. Gradient-based optimization or data assimilation (Barker et al., 2004), of course, is critically guided by gradients. In scenario analysis, parameters or inputs are perturbed, often beyond the observed ranges, to assess the responses. Without correctly capturing the sensitivity, the responses could be erroneous (Saltelli et al., 2019).

**Gradient Computation for PDE Solvers:** Gradient, or sensitivity computations, are widely used in the analysis of partial differential equations (PDEs). Traditionally, methods like finite differences can approximate gradients using existing numerical solvers and they remain relevant. Now we also have the option of using automatic differentiation (AD) offered by differentiable-programming platforms such as PyTorch, TensorFlow, Julia, or JAX, which enable efficient computation of gradients of outputs with respect to inputs. While the main purpose of implementing solvers on these platforms is often to train process-based equations together ("end-to-end") with NNs so that missing relationships can be learned along with partial knowledge (Innes et al., 2019; Shen et al., 2023; Zhu et al., 2023; Schoenholz & Cubuk, 2020; Tsai et al., 2021), these solvers can indeed provide rapid calculation of gradients as a byproduct. Gradients can be also computed by adjoint methods, either "discretize-then-optimize"(Onken & Ruthotto, 2020; Song et al., 2024a) or "optimize-then-discretize" (Chen et al., 2018). Differentiable PDE solvers are rapidly increasing across numerous domains, demonstrating highly competitive performance, especially in data-scarce (Feng et al., 2023) or unseen extreme (Song et al., 2024b) scenarios. Our work is compatible with either finite differences or AD. In this context, our work introduces a novel methodology that seamlessly integrates sensitivity analysis directly into the Fourier Neural Operator framework, significantly enhancing the operator's capability to tackle both forward and inverse problems. Notable studies such as Gradient enhanced neural networks Liu & Batill (2000) and Sobolev training Czarnecki et al. (2017) have explored the utilization of derivative information to improve neural network training, but they focused on low-dimensional approximation of derivatives. Our proposed Sensitivity-Constrained Fourier Neural Operator (SC-FNO) overcomes the problems with the estimation of parametric sensitivity, allowing better inversion while maintaining FNO-level computational efficiency for solving differential equations. The SC-FNO can provide highly accurate parameter sensitivities with less training data, ensuring that solution accuracy is maintained even under input perturbations and even concept drift (physical parameters in testing exceed ranges encountered during training). Although we focus attention on FNOs here, our experiments show the approach generalizes to many other neural operators (see **Appendix** D.1).

## 2 METHODOLOGY

Training data for Neural Operators are obtained by running physics-based model simulations multiple times with different initial conditions and physical parameters. We note that for differentiable physics-based models, the sensitivities $\partial \mathbf{u}/\partial \mathbf{p}$ can also be computed. For non-differentiable models, they can be approximated with finite differences. The training of SC-FNO uses a loss function $L_u$ over the solution paths (as with FNO) and also adds a loss $L_s$ over the parameter sensitivities. One can optionally add a PINN equation loss $L_{Eq}$ (resulting in SC-FNO-PINN). This framework can be used with other neural operators as well (see **Appendix** D.1).

### 2.1 SC-FNO: ENHANCING FNO WITH SENSITIVITY

**Original Fourier Neural Operators (Li et al., 2021):** FNOs represent a class of learning architectures designed to model mappings between infinite-dimensional function spaces, thereby providing efficient solutions to complex parametric PDEs. Mathematically, FNOs operate within a defined domain $\mathbf{D} \subset \mathbf{R}^d$. The function spaces, denoted as $\mathbf{A}(\mathbf{D}; \mathbf{R}^{d_a})$ for inputs and $\mathbf{U}(\mathbf{D}; \mathbf{R}^{d_u})$ for outputs, encapsulate the functional domains of the inputs and outputs respectively. The goal of an FNO is to approximate the operator $\mathbf{G} : \mathbf{A} \times \Theta \to \mathbf{U}$, which ideally maps each input function $\mathbf{a}_j$ in $\mathbf{A}$ to an output function $\mathbf{u}_j$ in $\mathbf{U}$. Formally, the FNO framework is expressed as:

$$\widetilde{\mathbf{G}} : \mathbf{A} \times \mathbf{\Theta} \to \mathbf{U}, \tag{1}$$

where $\mathbf{\Theta}$ represents the space of parameters that the FNO optimizes to reduce the discrepancy between predicted and true outputs. The Fourier transformation, essential in the operation of FNOs, is applied to transform function data into the frequency domain, facilitating the application of linear operators:

$$(\mathbf{F}\mathbf{f})_j(\mathbf{k}) = \int_{\mathbf{D}} \mathbf{f}_j(\mathbf{x}) e^{-2\pi i \langle \mathbf{x}, \mathbf{k} \rangle} \, \mathrm{d}\mathbf{x} \qquad \text{and} \qquad (\mathbf{F}^{-1}\mathbf{f})_j(\mathbf{x}) = \int_{\mathbf{D}} \mathbf{f}_j(\mathbf{k}) e^{2\pi i \langle \mathbf{x}, \mathbf{k} \rangle} \, \mathrm{d}\mathbf{k}. \tag{2}$$

This representation utilizes the convolution theorem to model complex nonlinear behaviors efficiently:

$$\mathbf{K}(\phi)\mathbf{v}_t(\mathbf{x}) = \mathbf{F}^{-1}(\mathbf{R}_\phi \cdot (\mathbf{F}\mathbf{v}_t))(\mathbf{x}), \tag{3}$$

where $\mathbf{R}_\phi$ represents a learnable parameterized function within the Fourier domain, allowing FNOs to encode and decode information effectively. While FNOs offer significant computational advantages, they necessitate extensive training data (Kovachki et al., 2021; Li et al., 2021). Moreover, generalizing beyond the training datasets remains a challenge.

**Our Contribution:** We introduce sensitivity-based loss terms to Neural Operators, resulting in a novel framework we call Sensitivity-Constrained Neural Operators (SC-NO) (Figure A.7 in **Appendix** A). For clarity, we demonstrate this concept primarily through its application to Fourier Neural Operators (FNO), yielding SC-FNO. However, we also show that the framework is generalizable to various types of neural operators, such as Wavelet Neural Operators (Tripura & Chakraborty, 2023), Multiwavelet Neural Operators (Gupta et al., 2021), and DeepONets (Wang et al., 2021) (descriptions and results are in **Appendix** D.1). The current neural operator training never harnessed sensitivity information (such as Jacobians) and never explicitly guaranteed how the inputs should be used in the model. However, our approach leverages either finite difference or a differentiable solver to compute these sensitivities. This development improves the robustness of prediction of both $\mathbf{u}$ and $\partial \mathbf{u}/\partial \mathbf{p}$ by ensuring that input variables are utilized correctly by the FNO — we dictate the sensitivity of the predicted outcomes with respect to input parameters and initial conditions, while in the future boundary conditions can also be incorporated. The model is expressed concisely as:

$$\mathbf{u}(\mathbf{x}, t) = \mathcal{F}_{\text{SC-FNO}}(\mathbf{u}_0, \mathbf{x}, t, \mathbf{p}), \tag{4}$$

where $\mathcal{F}_{\text{SC-FNO}}$ denotes the SC-FNO mapping, $\mathbf{u}_0$ are the initial conditions, $\mathbf{x}$ represents spatial coordinates, $t$ is time. The output $\mathbf{u}$ is governed by the differential equation:

$$\frac{\partial \mathbf{u}}{\partial t} = f(\mathbf{u}, \mathbf{x}, t, \mathbf{p}), \tag{5}$$

with $f$ defining the dynamics of the system. In this formulation, $\mathbf{p}$ encapsulates parameters that critically modulate, influence, or characterize the system, including physical constants, material

properties, or scale-dependent parameterizations. SC-FNO incorporates a sensitivity loss to enforce the accuracy of the predicted sensitivities (Jacobians) against those computed from the differentiable numerical solver. The SC-FNO model computes the solution **u** across all time and space in a single execution. The mathematical definition of sensitivity loss $L_s$ is as follows:

$$L_s = \frac{1}{M} \sum_{j=1}^{M} \left\| \frac{\partial \hat{\mathbf{u}}(\mathbf{x}_j, t_j; \mathbf{p})}{\partial \mathbf{p}} - \frac{\partial \mathbf{u}(\mathbf{x}_j, t_j; \mathbf{p})}{\partial \mathbf{p}} \right\|^2. \tag{6}$$

where $\mathbf{x}_j$ and $t_j$ represent the spatial and temporal coordinates of the sampled points at which the Jacobians are evaluated. $\partial \hat{\mathbf{u}}/\partial \mathbf{p}$ is the Jacobian of the predicted outputs with respect to the input parameters, obtained through AD applied to the SC-FNO. $\partial \mathbf{u}/\partial \mathbf{p}$ is the true Jacobian derived from precise differentiable numerical solvers or known analytical solutions. $M$ denotes the number of evaluation points across the domain, including points used to impose boundary and initial conditions.

## 2.2 THE PINN-LOSS EQUATION AS AN OPTIONAL REGULARIZER

Physics-Informed Neural Networks (PINNs) regularize the gradients of the neural networks using the governing differential equations at collocation points and boundary conditions Raissi et al. (2019); Karniadakis et al. (2021); He et al. (2022). The core PDE constraint in PINNs is represented as:

$$\mathcal{N}[\mathbf{u}(\mathbf{x}, t); \mathbf{p}] = 0$$

The PINN loss function, $L_{Eq}$, incorporates the discrepancy from the PDE with terms for initial and boundary conditions ($L_{IC}$ and $L_{BC}$):

$$L_{Eq} = L_{PDE} + \alpha(L_{IC} + L_{BC}) = \frac{1}{N} \sum_{i=1}^{N} |\mathcal{N}[\mathbf{u}(\mathbf{x}_i, t_i); \mathbf{p}]|^2 + \alpha(L_{IC} + L_{BC}),$$

where $\alpha$ is a weighting factor, and $N$ denotes the number of collocation points in the computational domain. PINNs trained for specific conditions often cannot predict general evolutionary trajectories or guarantee accurate parameter sensitivity Jin et al. (2021); Ren et al. (2022). Additionally, PINNs can be computationally slower than traditional numerical solvers. In this work, we evaluate the ability of PINN-type loss to improve prediction accuracy along with sensitivities. In a broadened sense, SC-FNO can be considered a novel type of PINN as it also regularizes the gradient of the neural networks, but their procedures are fundamentally different. PINNs do not have access to this sensitivity, as usual PDEs do not contain $\partial \mathbf{u}/\partial \mathbf{p}$. SC-FNO prepares gradient data to supervise $\partial \mathbf{u}/\partial \mathbf{p}$ but do not require optimization during forward simulation, whereas PINNs rely on collocation points for equation-based loss optimization to supervise ($\partial \mathbf{u}/\partial \mathbf{x}$, $\partial \mathbf{u}/\partial \mathbf{t}$), but not $\partial \mathbf{u}/\partial \mathbf{p}$. SC-FNO is designed to serve as an efficient forward simulator that can handle various changes in high dimensional input, which can also help with inversion tasks.

## 2.3 GRADIENT COMPUTATION METHODS FOR DIFFERENTIAL EQUATIONS

To prepare training and validation datasets containing true solution paths along with their sensitivity (gradients with respect to parameters), we developed and implemented two distinct approaches:

1. A differentiable numerical solver based on the `torchdiffeq` package. We extended this ODE-oriented framework to handle PDEs by reformulating them in the form $\frac{d\mathbf{u}}{dt} = \text{RHS}(\mathbf{x})$, where $\text{RHS}(\mathbf{x})$ encapsulates terms related to spatial derivatives. This solver leverages PyTorch's AD capability, although adjoint methods can also be used. The computation time for preparing datasets with and without Jacobian computation is presented in Table D.12.

2. Finite difference methods for gradient computation with a traditional solver. The method approximates gradients by solving the PDE multiple times with slightly perturbed parameter values and computing the differences between the solutions. This process offers a non-intrusive means of gradient estimation for any existing numerical solvers.

We implemented efficient functions to compute the sensitivities. These solution paths and sensitivities are computed once and archived for later use, though they could also be generated on the fly if desired. This represents a one-time cost per equation, and since input impacts are resolved upfront, no repeated preparation is needed for different parameters. We assessed the effectiveness of both gradient computation methods and compared their accuracies.

### 2.4 IMPLEMENTATION DETAILS

The SC-FNO architecture processes parameters ($\mathbf{p}$) alongside spatial coordinates and initial conditions through the lifting layer as function inputs. This layer reshapes and repeats parameters to match the problem's spatial-temporal dimensions, then concatenates them with other inputs before neural network processing. Notably, both FNO and SC-FNO share identical neural network architectures and inputs, differing only in their loss configurations. After computing true gradients ($\partial \mathbf{u}/\partial \mathbf{p}$) once during dataset preparation (Section 2.3), our training procedure contains several efficiency-focused steps. Instead of computing gradients at all points, we randomly select a subset of spatial-temporal points in each epoch ($n < N$ spatial points $\times$ $t < T$ time points) where $n < N$ and $t < T$. The neural operator's predicted gradients at these points are computed using AD and compared with the pre-computed true gradients. This sampling varies between epochs to eventually cover the full solution space. This procedure eliminates the need for additional solver runs during training while maintaining effective sensitivity supervision. Each minibatch requires only one forward pass before applying AD, adding minimal computational overhead. The pre-computed gradients are stored and reused throughout training, making the approach efficient as parameter dimensionality increases.

## 3 EXPERIMENTS

We evaluated four different configurations of the FNO models, each defined by a unique combination of loss functions. The first configuration uses only the data loss $L_\mathbf{u}$ (FNO), which matches predicted state variables with the solution paths; The second, $L_\mathbf{u} + L_{\text{Eq}}$ (FNO-PINN), incorporates the equation loss; The third, $L_\mathbf{u} + L_{\text{Eq}} + L_s$ (SC-FNO-PINN), combining both sensitivity and equation losses; and finally, $L_\mathbf{u} + L_s$ (SC-FNO), which adds only sensitivity loss to data loss. Pseudocodes for these models can be found in the **Appendix A**. We evaluated the competing methods on the following well-known differential equations.

**ODE1: Composite Harmonic Oscillator** and **ODE2: Duffing Oscillator Equation** each operate within a temporal domain of $t \in [0, 1]$, where was discretized into $N = 100$ equal time steps. Our goal was to learn the operator that maps the first $M$ time steps of solutions $u$, alongside parameters $\mathbf{p}$, to the solutions at the next $N - M$ subsequent time steps, formulated as $u : [0 : M] \cup \mathbf{p} \to u : [M : N]$.

**PDE1: Generalized Nonlinear Damped Wave Equation** is explored within both a temporal domain $t \in [0, 1]$ and a spatial domain $x \in [0, 1]$. The temporal domain was discretized into $N = 30$ equal time steps and the spatial domain was discretized into $S_x = 20$. We aim to learn the operator that maps from the first $M$ initial time steps of $u$, alongside parameters $\mathbf{p}$, to the next $N - M$ subsequent time steps, expressed as $u : [0, S_x]^1 \times [0, M] \cup \mathbf{p} \to u : [0, S_x]^1 \times [M, N]$.

**PDE2: Forced Burgers' Equation** is analyzed across a time interval $t \in [0, \pi]$ and a spatial domain $x \in [0, 1]$. The temporal domain is segmented into $N = 30$ equal parts, and the spatial domain into $S_x = 40$ parts. The aim is to identify the operator that maps the first $M$ time steps of $u$, in conjunction with parameters $\mathbf{p}$, to the remaining $N - M$ time steps, formulated as $u : [0, S_x]^1 \times [0, M] \cup \mathbf{p} \to u : [0, S_x]^1 \times [M, N]$.

**PDE3: Stream Function-Vorticity Formulation of the Navier-Stokes Equations** spans the temporal domain $t \in [0, 3]$ and spatial domains $x, y \in [0, 1]$. The spatial domain and the temporal domain are $[0, 1]$ and $[0, 3]$, respectively. The 2D spatial domain was discretized into $S_x = S_y = 64$ equal spatial divisions. Here, our goal is to learn an operator that takes initial conditions of $u$ along with parameters $\mathbf{p}$, and directly maps them to the solution of vorticity at the final time step $t = 3s$, formulated as $u : [0, S_x] \times [0, S_y] \times [t = 0(s)] \cup \mathbf{p} \to u : [0, S_x] \times [0, S_y] \times [t = 3(s)]$.

**PDE4: Allen-Cahn equation** is analyzed within temporal domain $t \in [0, 1.0]$ and spatial domain $x \in [0, 1]$, discretized into $N = 30$ and $S_x = 40$ parts respectively. We aim to learn the operator that maps from first $M$ time steps of $u$ and parameters $\mathbf{p}$ to the next $N - M$ time steps, formulated as $u : [0, S_x]^1 \times [0, M] \cup \mathbf{p} \to u : [0, S_x]^1 \times [M, N]$.

Detailed specifications of the differential equations and the ranges of their parameters are provided in Table B.6 in **Appendix B**. The architectural details, hyperparameters and training time information are comprehensively presented in Tables C.7 and C.8 in **Appendix C**.

In the following, we first demonstrate the superior performance of SC-FNO in inversion (or optimization) tasks compared to FNO alone. Then we explain such outperformance using several experiments: we show how trained FNOs alone captured the gradients poorly and thus fared worse than SC-FNOs when inputs are perturbed, which often occurs during inversion. Then, we show

that SC-FNO requires fewer training samples to reach higher quality, particularly when the input dimension is higher. Finally, we show that finite difference can also be functional. Experiments were run with various neural operators and PDEs to ensure the generality of the conclusions.

## 3.1 PARAMETER INVERSION FROM SOLUTION PATHS

Parameter inversion with PDEs plays a pivotal role in system identification, calibration, and optimization, and serves as a primary area benefiting from the efficiency of surrogate models. Our inversion experiments utilized FNO and SC-FNO surrogate models, which were trained and rigorously evaluated on synthetic datasets as described in Section 3.2 for PDE1-PDE3. These datasets, containing $2 \times 10^3$ training samples, were generated using the differentiable numerical solver. They feature randomly generated initial conditions and parameter values, alongside their corresponding solutions at multiple time points (Table B.6 in **Appendix** B) and Jacobians. We used 70% of the data for training, 15% for validation, and 15% for testing, and ensured that the validation and test sets contained parameter values not encountered during training. The first experiment inverted the models to infer the $\alpha$ parameter alone while treating others as known. We then used backpropagation to optimize the parameter by minimizing the discrepancy between the synthetic data and PDE solutions. The second task simultaneously inverted all parameters of either PDE1 or PDE2. SC-FNO demonstrates notably superior performance over FNO and FNO-PINN in simple inversion tasks, achieving nearly perfect inversion while the latter two showed significantly more scattering (Figure 1). In the single-parameter inversion, SC-FNO ($R^2 = 0.998$) achieves less than 1/5 and 1/4 the relative L² inversion errors of FNO ($R^2 = 0.905$) and FNO-PINN, respectively (Figure 1a). Multi-parameter inversion incurs larger uncertainty and greater contrasts — SC-FNO has 1/6 and 1/2.8 the relative L² inversion errors of FNO and FNO-PINN, respectively (Figures 1b and 2). Retrieving $\alpha$ in the multi-parameter case results in large heteroscedastic scattering with FNO ($R^2 = 0.635$), whereas SC-FNO remains highly accurate (Figure 1b). For PDE2 with four parameters, SC-FNO's $R^2$ values are above 0.96, while those of FNO hover around 0.85. For PDE1 with five parameters, SC-FNO maintains $R^2$ above 0.94 for all parameters, while those of FNO drop below 0.64, suggesting overfitting and a breakdown of the surrogate model. Similar contrasts are found with PDE3 (Navier Stokes) (Figure D.10 in **Appendix** D). This performance gap may widen further with increased parameter dimensions. FNO-PINN's relative L² is half of that of FNO (Figures 1 and 2) but still 3x-5x that of SC-FNO. Additional scatter plots for parameter inversion are available in Figures D.8 and D.9 in **Appendix** D. The introduction of the sensitivity loss lead to consistent enhancements across other neural operators like WNO, MWNO, and DeepONet, with uniform conclusions reflected in Table D.11 in **Appendix** D. These enhancements are larger than the differences between different neural operators. By training on both solution data and their gradients, SC-FNO builds a more robust internal representation of the PDE dynamics, particularly regarding the roles of inputs. The next sections explore in more depth the advantages of gradient-aware surrogate models.

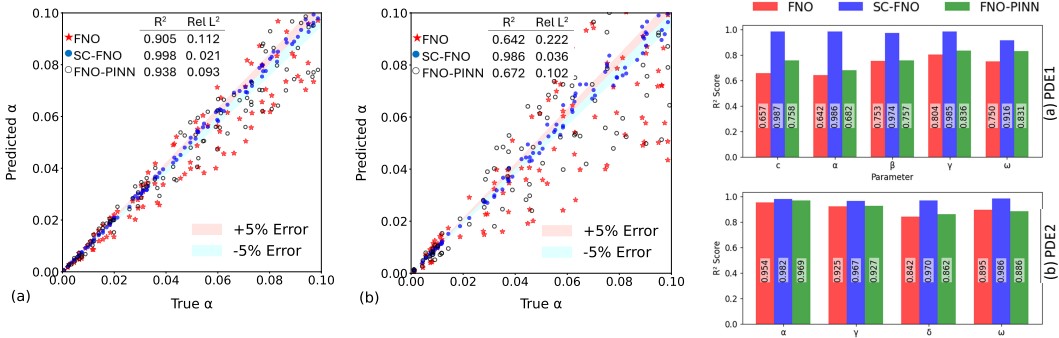

Figure 1: Inversion of the parameter $\alpha$ in PDE1 using FNO and SC-FNO models (a) single parameter inversion, (b) simultaneous multi-parameter inversion.

Figure 2: Simultaneous multi-parameter inversion accuracy for PDEs 1 and 2 using FNO and SC-FNO.

## 3.2 SURROGATE MODEL QUALITY AND ROBUSTNESS TO INPUT PERTURBATION

Surrogate quality assessments indicate that the superior inversion performance of SC-FNO, even with so few parameters, can be attributed to the model's unique ability to accurately capture parameter sensitivities and its robustness to input perturbations. As discussed earlier, the surrogate models were trained on identical input and output datasets uniformly prepared for each differential equation, but only SC-FNO or SC-FNO-PINN used parameter sensitivities. We first evaluated the models on test datasets with parameters ranging from [a, b], the same as the training data, as detailed in Table B.6 in **Appendix** B. Subsequently, to assess the models' generalizability, we perturbed the parameters beyond their original ranges by applying various perturbation percentages, $\lambda$, resulting in a parameter range of ([b, (1+$\lambda$)b]). By systematically increasing $\lambda$, we evaluated the models' performance at various levels of extrapolation beyond the training dataset.

Table 1: Error metrics for PDE1 (5 parameters) and PDE2 (4 parameters) with $2 \times 10^3$ training samples. Both have low dimensional parameters.

(a) PDE1

| Value | Metric | Original range of parameters | | | | Perturbed range ($\lambda : 0.4$) | | | |
|---|---|---|---|---|---|---|---|---|---|
| | | FNO | SC-FNO | SC-FNO-PINN | FNO-PINN | FNO | SC-FNO | SC-FNO-PINN | FNO-PINN |
| $u(t)$ | R² | 0.986 | 0.983 | **0.989** | 0.978 | 0.529 | 0.912 | 0.928 | 0.620 |
| | Relative L² | 0.0146 | 0.0175 | **0.0112** | 0.0220 | 0.4716 | 0.0882 | 0.0721 | 0.3805 |
| $\frac{\partial u}{\partial c}$ | R² | 0.723 | 0.924 | **0.943** | 0.801 | 0.515 | 0.901 | 0.915 | 0.605 |
| | Relative L² | 0.2772 | 0.0761 | **0.0577** | 0.1992 | 0.4855 | 0.0993 | 0.0851 | 0.3953 |
| $\frac{\partial u}{\partial \alpha}$ | R² | 0.741 | 0.925 | **0.945** | 0.805 | 0.522 | 0.908 | 0.924 | 0.615 |
| | Relative L² | 0.2590 | 0.0750 | **0.0552** | 0.1955 | 0.4780 | 0.0924 | 0.0767 | 0.3852 |
| $\frac{\partial u}{\partial \beta}$ | R² | 0.763 | 0.930 | **0.956** | 0.817 | 0.530 | 0.914 | 0.930 | 0.622 |
| | Relative L² | 0.2370 | 0.0700 | **0.0441** | 0.1834 | 0.4708 | 0.0862 | 0.0732 | 0.3781 |
| $\frac{\partial u}{\partial \gamma}$ | R² | 0.772 | 0.931 | **0.955** | 0.815 | 0.539 | 0.920 | 0.936 | 0.630 |
| | Relative L² | 0.2283 | 0.0696 | **0.0451** | 0.1858 | 0.4610 | 0.0801 | 0.0644 | 0.3706 |
| $\frac{\partial u}{\partial \omega}$ | R² | 0.781 | 0.932 | **0.963** | 0.825 | 0.545 | 0.924 | 0.937 | 0.632 |
| | Relative L² | 0.2190 | 0.0682 | **0.0375** | 0.1752 | 0.4557 | 0.0761 | 0.0633 | 0.3687 |

(b) PDE2

| Value | Metric | Original range of parameters | | | | Perturbed range ($\lambda : 0.4$) | | | |
|---|---|---|---|---|---|---|---|---|---|
| | | FNO | SC-FNO | SC-FNO-PINN | FNO-PINN | FNO | SC-FNO | SC-FNO-PINN | FNO-PINN |
| $u(t)$ | R² | **0.997** | **0.997** | 0.995 | 0.995 | 0.734 | 0.933 | 0.923 | 0.802 |
| | Relative L² | 0.0029 | **0.0016** | 0.0065 | 0.0073 | 0.0325 | 0.0112 | 0.0124 | 0.0287 |
| $\frac{\partial u}{\partial \alpha}$ | R² | 0.206 | **0.987** | 0.907 | 0.137 | 0.152 | 0.904 | 0.830 | 0.113 |
| | Relative L² | 0.2092 | **0.0135** | 0.0813 | 0.8545 | 1.1236 | 0.0755 | 0.0987 | 0.9865 |
| $\frac{\partial u}{\partial \gamma}$ | R² | 0.423 | **0.986** | 0.991 | 0.519 | 0.311 | 0.903 | 0.904 | 0.429 |
| | Relative L² | 0.8542 | **0.0540** | 0.0523 | 0.7566 | 0.9875 | 0.0767 | 0.0755 | 0.8244 |
| $\frac{\partial u}{\partial \delta}$ | R² | 0.821 | 0.912 | **0.957** | 0.871 | 0.604 | 0.835 | 0.876 | 0.719 |
| | Relative L² | 0.1245 | 0.0756 | **0.0567** | 0.1023 | 0.2544 | 0.1023 | 0.0888 | 0.1567 |
| $\frac{\partial u}{\partial \omega}$ | R² | 0.321 | **0.982** | 0.912 | 0.427 | 0.236 | 0.912 | 0.834 | 0.353 |
| | Relative L² | 0.8856 | **0.0567** | 0.0789 | 0.7899 | 1.0244 | 0.0722 | 0.0944 | 0.8878 |

For PDEs, when the test data originate from the same parameter ranges as the training data (Table B.6 in **Appendix** B), the solution paths are of high quality, but FNO learns sensitivities significantly poorer than those of SC-FNO (Table 1 left half). While the metrics for $u$ are similarly high among all models, the $R^2$ values for FNO gradients range only from 0.72 to 0.78 (relative L²: 0.28 to 0.22) for PDE1 and from 0.21 to 0.82 (relative L²: 0.21 to 0.12) for PDE2. In contrast, SC-FNO and SC-FNO-PINN consistently achieve $R^2$ values of 0.92-0.93 (relative L²: 0.08-0.07) for PDE1 and 0.91-0.99 (relative L²: 0.09-0.1) for PDE2. The inclusion of a PINN-type equation loss ($L_{Eq}$) in FNO-PINN provides only minor benefits for $\frac{\partial \mathbf{u}}{\partial \mathbf{p}}$, with $R^2$ values remaining below 0.52 (relative L²: >0.48) for most gradients in PDE2. These improvements are not comparable to those achieved by SC-FNO and SC-FNO-PINN, highlighting the predominant impact of $L_s$ terms. ODEs show a similar pattern where SC-FNO has much better sensitivity accuracy for both parameters (**Appendix** Table D.14) and initial conditions ($\gamma$ in Table D.14a and $\zeta$ in Table D.14b). In some sample test predictions (Figure 3a-d), surprisingly large and unphysical oscillations are observed in FNO-predicted sensitivities. These simple cases were designed to show that a surrogate model accurately predicting $u$ may fail to capture the dependence of solution paths on the inputs, leading to large inversion errors.

This pattern is reliably repeated for more challenging PDEs with a small number of parameters, e.g., for PDE3 (Navier Stokes, 2 parameters), where models lacking the sensitivity loss well predicts

Table 2: Error metrics for PDE3 with $1 \times 10^3$ training Samples. Both have

| | $\omega$ | | $\partial\omega/\partial\alpha$ | | $\partial\omega/\partial\beta$ | |
|---|---|---|---|---|---|---|
| Method | Relative L² | $R^2$ | Relative L² | $R^2$ | Relative L² | $R^2$ |
| FNO | 0.0312 | **0.997** | 0.7230 | 0.036 | 0.9642 | 0.036 |
| SC-FNO | 0.0345 | 0.994 | 0.0112 | **0.986** | 0.0132 | **0.987** |

vorticity but falter in capturing the sensitivities (Table 2), missing both fine-scale features and large-scale patterns (Figure 6). In contrast, SC-FNO accurately recreates patterns and even fine-scale details. In the case of PDE4 (Allen-Cahn), a challenging test case due to its bifurcation nature where small parameter changes can cause abrupt phase transitions in solutions, SC-FNO exhibits mildly better $u$ solution accuracy than FNO (Table 3). More noticeably, SC-FNO generates 1/25 the Jacobian error as FNO, even with reduced samples near critical parameter values where solution behavior changes sharply. Given the diverse dynamics in the equations tested, the general incapability of FNO to capture sensitivity is evident. Note that SC-FNO does not entail excessive additional training cost (Table C.8). We also tested a broad range of other neural operators, all of which exhibited markedly improved sensitivities upon introducing the sensitivity loss (**Appendix** Table D.9). The difficulty FNO has in capturing parameter gradients leads to prominently degraded performance under input perturbations, though SC-FNO remains robust and reliable. The $R^2$ for the solution **u** at 40% perturbation decreases precipitously to 0.529 (relative L²: 0.471) for PDE1 and 0.734 (relative L²: 0.266) for PDE2, compared to 0.912 (relative L²: 0.088, 1/5 that of FNO's) and 0.933 (relative L²: 0.067, 1/4 that of FNO's) for SC-FNO, respectively (Table 1). As the perturbation ratio increases for PDE1, a stark decline in $R^2$ is observed for FNO, in contrast to the relatively stable SC-FNO (Figure 5). We argue that the large perturbation-induced error is a primary reason for FNO's poor performance in inversion tasks, during which the search algorithm ventures into under-sampled regions of the parameter space, consistent with the arguments in Li et al. (2023). At an accuracy level of $R^2 = 0.529$ (relative L²: 0.471) (as illustrated in Figure 1b by the scatter plot), the surrogate model loses its ability to guide the inversion. Even without straying beyond the training parameter range, a mere change in the pattern of inter-parameter correlations could cause a departure from the training conditions (concept drift). While increasing training data can help, it becomes exponentially more difficult to adequately cover the parameter space for FNO as the input dimension increases. This challenge is highlighted by the contrasts observed between Figures 1a and 1b, as well as between Figures 2a and 2b. Additionally, SC-FNO-PINN only shows a slight benefit in L² over SC-FNO in all of these tests (Table 1). The minimal impact of the equation loss is both surprising and previously unexamined. This ineffectiveness likely stems from the absence of terms related to (time-integrated) $\partial\mathbf{u}/\partial\mathbf{p}$ in the differential equations, which leaves this sensitivity unconstrained even though **p** is in the equation. For coupled optimization tasks, it is imperative that surrogate models not only reflect the physical phenomena accurately but also adapt to new, unobserved conditions. SC-FNO thus provides an effective alternative solution.

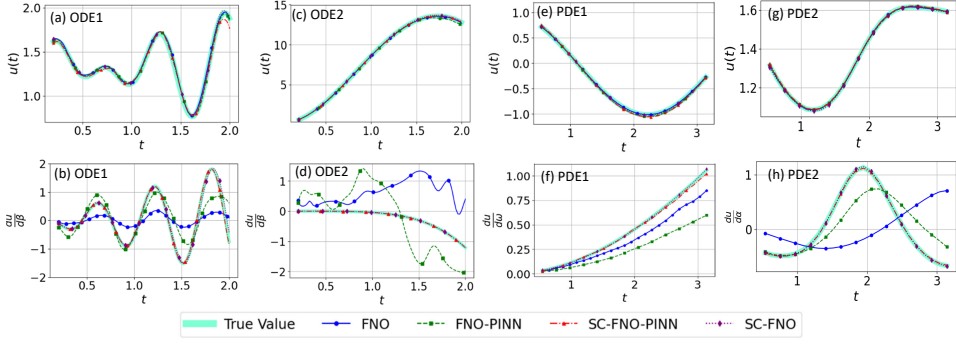

Figure 3: Sample prediction of models for ODEs and PDE1 and PDE2.

## 3.3 MODEL PERFORMANCE ACROSS VARYING TRAINING DATA VOLUMES

We explored how different models respond to varying amounts of training data. Specifically, we aimed to investigate how the integration of different loss terms affects both the accuracy and the

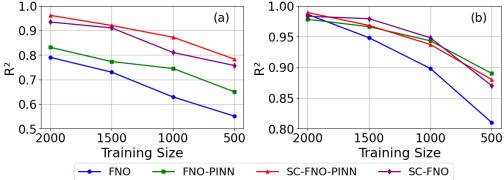

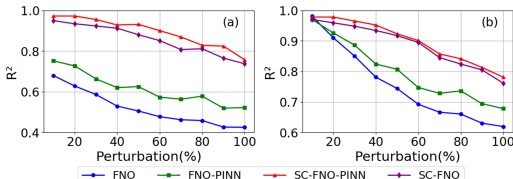

Figure 4: Models' performance on PDE1 across training sample sizes, (a) $\frac{\partial u}{\partial \omega}$ (b) $u(t)$.

Figure 5: Performance of models for PDE1 for perturbed datasets, (a) Standardized $\frac{\partial \hat{u}}{\partial \mathbf{p}}$ and (b) $u(t)$.

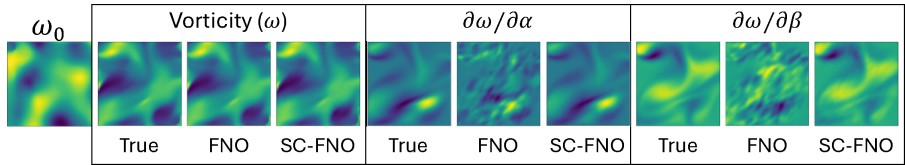

Figure 6: Sample prediction of models for PDE 3

generalization capabilities of the models under limited training data scenarios. The remaining portion of the dataset, not used for training, served to test and measure the models' performance.

Table 3: Performance comparison for PDE4

| Metrics | N=500 | | N=100 | |
|---|---|---|---|---|
| | FNO | SC-FNO | FNO | SC-FNO |
| *State Value Metrics* | | | | |
| R² | 0.999 | 0.999 | 0.997 | 0.998 |
| Relative L² | 0.0110 | 0.0081 | 0.0205 | 0.0151 |
| *Mean Jacobian Metrics* | | | | |
| R² | -3.11 | 0.998 | -5.8373 | 0.993 |
| Relative L² | 0.5212 | 0.0207 | 0.5830 | 0.0486 |

Table 4: Performance comparison for zoned PDE2

| Metrics | N=500 | | N=100 | |
|---|---|---|---|---|
| | FNO | SC-FNO | FNO | SC-FNO |
| *State Value Metrics* | | | | |
| R² | 0.960 | 0.997 | 0.927 | 0.996 |
| Relative L² | 0.0282 | 0.0073 | 0.0387 | 0.0087 |
| *Mean Jacobian Metrics* | | | | |
| R² | -8.332 | 0.949 | -14.012 | 0.927 |
| Relative L² | 1.9627 | 0.1770 | 2.4623 | 0.2134 |

The experiment demonstrates that FNO performance can rapidly degrade as the volume of training data decreases, whereas SC-FNO can maintain higher accuracy and continue to function effectively as a surrogate model (Figure 4). While all models exhibit a decrease in $R^2$ scores for both state values $u(t)$ and gradients $\frac{\partial \hat{u}}{\partial \mathbf{p}}$ as training sizes diminish, SC-FNO and SC-FNO-PINN show a notably slower rate of decline. With only 500 training samples, FNO's $R^2$ dropped to 0.8, displaying an acceleration in degradation, while SC-FNO maintained an $R^2$ around 0.9, comparable to FNO's performance with 1,000 training samples. This advantage is attributed to the use of higher-order (gradient) information, which, similar to traditional numerical methods such as spline interpolation or second-order finite differences, typically results in a better rate of error convergence. This sensitivity to the volume of training data contributed to FNO's poor performance in inversion tasks (Section 3.1), indicating that dense training data is necessary throughout the parameter search space to ensure accuracy. In practical applications where comprehensive datasets are unattainable, the ability to maintain high model performance with fewer examples is invaluable.

## 3.4 HANDLING FUNCTIONAL AND HIGH-DIMENSIONAL PARAMETER SPACE

All previous examples have small numbers of scalar parameters. To evaluate the models' effectiveness with higher parameter dimensions, we modified PDE2 (Burger's equation) with zoned parameters. We divided the spatial domain into $S = 40$ segments with different advection $\alpha$ and forcing amplitude $\delta$ in each zone, along with global parameters $\gamma$ and $\omega$, resulting in $2S + 2 = 82$ total parameters. Maintaining the settings from Section 3.2, models were trained with different sample sizes. SC-FNO's advantages become immediately prominent from this higher dimensional case, even for the solution path **u** itself. At 500 samples, SC-FNO has only 1/4 the relative L² error (0.0073) as FNO (0.0282, Table 4) while requiring moderately more training time (Appendix Table C.8). With $N = 100$ samples, SC-FNO maintains the low error (relative L² = 0.0087) while FNO degrades significantly (relative L² = 0.039). In fact, SC-FNO with 100 samples has less than 1/3 of the relative L² error of FNO with 500 samples, and also requires less time to train. SC-FNO's level of accuracy seems

elusive for FNO, which sees L² error decreasing by only 28% as $N$ increases from 100 to 500, and thus we argue SC-FNO lifts the performance ceiling. The contrast is more dramatic for Jacobian predictions (SC-FNO: relative L² = 0.213; FNO: relative L² = 2.46), which, as discussed earlier, has strong implications for the success of inversion and generalizability. These results highlight SC-FNO's capability to handle high-dimensional parameter spaces efficiently, maintaining accuracy even with limited training data. We can reasonably hypothesize that drastically increasing the number of parameters will further highlight SC-FNO's unique capability, which we reserve for future work.

## 3.5 DIFFERENT GRADIENT CALCULATION METHODS

We tested gradients derived from both a differentiable solver with automatic differentiation (AD) and a fourth-order finite difference solver (FD) using a traditional solver against the analytical solution for gradients of ODE1. The results of this verification are presented in Table D.13 in **Appendix D.3**. Following this validation, we used these methods to generate solution paths and sensitivities for training and testing surrogate models. We trained SC-FNOs using 70% of the produced data, dividing the remainder into 15% each for validation and testing. The validation and testing datasets included parameter values not featured in the training data, testing the models' generalization capabilities. SC-FNOs trained with either AD- or FD-generated gradients proved effective, producing accurate predictions for both solutions and their gradients, as shown in Table 5. These models both achieved R² > 0.95 (relative L² < 0.05) for solution paths and R² > 0.9 (relative L² < 0.1) for sensitivities. While AD provides higher accuracy and efficiency (Table D.13), FD remains effective and applicable to any existing model code. This makes SC-FNO versatile while maintaining computational efficiency comparable to traditional methods.

Table 5: Performance comparison of SC-FNO and SC-FNO-PINN using AD and FD solvers.

| Value | Metric | ODE1 | | | | PDE1 | | | |
|---|---|---|---|---|---|---|---|---|---|
| | | SC-FNO (AD) | SC-FNO (FD) | SC-FNO-PINN(AD) | SC-FNO-PINN(FD) | SC-FNO (AD) | SC-FNO (FD) | SC-FNO-PINN(AD) | SC-FNO-PINN(FD) |
| $u$ | R² | 0.991 | 0.968 | 0.994 | 0.983 | 0.983 | 0.963 | 0.989 | 0.978 |
| | Relative L² | 0.009 | 0.032 | 0.006 | 0.017 | 0.017 | 0.037 | 0.011 | 0.022 |
| Avg. $\frac{\partial u}{\partial \mathbf{P}}$ | R² | 0.996 | 0.987 | 0.995 | 0.983 | 0.928 | 0.913 | 0.952 | 0.932 |
| | Relative L² | 0.004 | 0.013 | 0.005 | 0.017 | 0.072 | 0.087 | 0.048 | 0.068 |

## 3.6 FURTHER DISCUSSION AND CONCLUSION

This work has demonstrated the powerful regularizing effect of sensitivities. The effectiveness of SC-FNO stems from the explicit governance of input influence through time-integrated parameter sensitivities ($\partial u / \partial p$). Unlike PINNs, which supervise spatial-temporal derivatives ($\partial u / \partial x$, $\partial u / \partial t$) through equation-based loss optimization, SC-FNO directly constrains parameter sensitivities typically absent in PDE formulations, using forward numerical models with differentiable solvers to prepare sensitivity data. This fundamental difference enhances model interpretability and reliability, making SC-FNO more suitable for coupled inversion or optimization tasks. Especially when input parameters have a higher dimension, SC-FNO can reduce training data demand, maintain robustness and generalizability, elevate the performance ceiling of neural operators, and even reduce training time. Our innovations include employing differentiable numerical solvers—and alternatively, finite differences—for computing (almost) exact gradients. Although differentiable programming is gaining traction in various domains (Shen et al., 2023; Song et al., 2024a; Aboelyazeed et al., 2023), few studies have leveraged computed gradients beyond backpropagation. We have used these gradients to supervise FNOs, calculating second-order gradients during training. The additional computational cost remains affordable—training the FNO on PDE1 used 722 MB while training SC-FNO required 764 MB. We argue that sensitivity regularization and time-step-free methods like FNO complement each other exceptionally well. SC-FNO's programmatic differentiability allows its seamless integration with neural networks (NNs), speeding up hybrid NN-based learning and optimization. Accurate gradients are crucial for the effective training of coupled NNs, thus opening up new avenues in the field of AI-enhanced solutions to differential equations.

ACKNOWLEDGEMENTS

This work was primarily supported by subaward A23-0249-S001 from the Cooperative Institute for Research to Operations in Hydrology (CIROH) through the National Oceanic and Atmospheric Administration (NOAA) Cooperative Agreement (grant no. NA22NWS4320003). The statements, findings, conclusions, and recommendations are those of the authors and do not necessarily reflect the view of NOAA. It was also partially supported by the U.S. Department of Energy, Office of Science, under award no. DE-SC0021979. Chaopeng Shen has financial interests in HydroSapient, Inc., a company that could potentially benefit from the results of this research. This interest has been reviewed by The Pennsylvania State University in accordance with its conflict of interest policy to ensure the objectivity and integrity of the research. The other authors have no competing interests to declare.

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

APPENDIX

## A    SC-FNO ARCHITECTURE AND SENSITIVITY INTEGRATION

Figure A.7 illustrates the schematic of the SC-FNO model architecture. In this framework, this variation of the Fourier Neural Operator (FNO) integrates various inputs: parameters ($\mathbf{P}$) that influence the differential equation, spatial and temporal coordinates ($X : [x, y, t]$), and the function $a(x)$, which may represent different initial conditions. This setup enables comprehensive learning and adaptation across different scenarios by leveraging both automatic differentiation for optimization and multiple loss components tailored to the specific dynamics and constraints of the system. Furthermore, in our framework, the FNO can be regularized by both a PINN-style differential equation ($L_{Eq}$) as well as the sensitivities (gradients of the solution path of a differential equation with respect to parameters contributing to the diff equation). These regularization terms ensure that the model adheres to the underlying physical laws that reflect how the solution path is determined by a parameter and how it changes when a parameter changes.

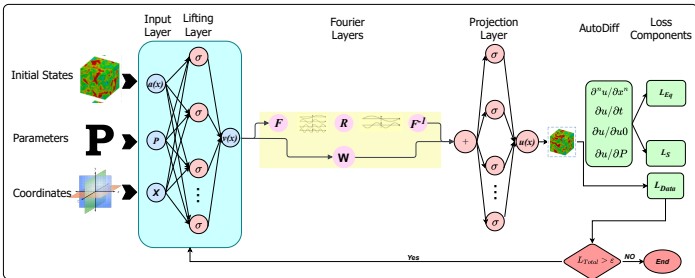

Figure A.7: Schematic of SC-FNO architecture.

Pseudocode for different FNO models, each with various loss configuration settings, are presented as follows. This section aims to highlight how different loss functions can be integrated and optimized within these models to enhance their predictive accuracy and performance.

---

**Algorithm 1** FNO Training Loop with $L_u$ Loss Over Epochs

---

 1: Initialize FNO model
 2: **for** epoch $= 1$ to max_epochs **do**
 3:     Shuffle training data
 4:     **for** each batch $\mathbf{P}$, $\mathbf{u}_{\text{true}}$ in training data **do**
 5:         Predict state values using FNO: $\hat{\mathbf{u}} \leftarrow \text{FNO}(\mathbf{P})$
 6:         Calculate loss: $L_u = \text{loss}(\hat{\mathbf{u}}, \mathbf{u}_{\text{true}})$
 7:         Backpropagate loss and update FNO model
 8:     **end for**
 9:     Evaluate on validation set
10:      Record training and validation loss
11: **end for**

---

---

**Algorithm 2** SC-FNO Training Loop with $L_u$ and $L_s$ Loss Over Epochs

---

1:  Initialize FNO model
2:  **for** epoch = 1 to max_epochs **do**
3:      Shuffle training data
4:      **for** each batch $\mathbf{P}$, $\mathbf{u}_{\text{true}}$, $\frac{\partial \mathbf{u}_{\text{true}}}{\partial \mathbf{P}}$ in training data **do**
5:          Predict state values using FNO: $\hat{\mathbf{u}} \leftarrow \text{FNO}(\mathbf{P})$
6:          Calculate primary loss: $L_u = \text{loss}(\hat{\mathbf{u}}, \mathbf{u}_{\text{true}})$
7:          Predict Jacobian of state values using Auto Diff (AD): $\hat{\mathbf{J}} \leftarrow \frac{\partial \hat{\mathbf{u}}}{\partial \mathbf{P}}$
8:          Calculate sensitivity loss: $L_s = \text{loss}(\hat{\mathbf{J}}, \frac{\partial \mathbf{u}_{\text{true}}}{\partial \mathbf{P}})$
9:          Calculate total loss: $L_{\text{total}} = c_1 \cdot L_u + c_2 \cdot L_s$
10:         Backpropagate total loss and update FNO model
11:     **end for**
12:     Eevaluate on validation set
13:     Record training and validation loss
14: **end for**

$\triangleright$ $c_1$ and $c_2$ are learnable coefficients

---

**Algorithm 3** SC-FNO-PINN Training Loop with $L_u$, $L_s$, and $L_{eq}$ Loss Over Epochs

---

1:  Initialize FNO model
2:  **for** epoch = 1 to max_epochs **do**
3:      Shuffle training data
4:      **for** each batch $\mathbf{P}$, $\mathbf{u}_{\text{true}}$, $\frac{\partial \mathbf{u}_{\text{true}}}{\partial \mathbf{P}}$ in training data **do**
5:          Predict state values using FNO: $\hat{\mathbf{u}} \leftarrow \text{FNO}(\mathbf{P})$
6:          Calculate primary loss: $L_u = \text{loss}(\hat{\mathbf{u}}, \mathbf{u}_{\text{true}})$
7:          Predict Jacobian of state values using Auto Diff (AD): $\hat{\mathbf{J}} \leftarrow \frac{\partial \hat{\mathbf{u}}}{\partial \mathbf{P}}$
8:          Calculate sensitivity loss: $L_s = \text{loss}(\hat{\mathbf{J}}, \frac{\partial \mathbf{u}_{\text{true}}}{\partial \mathbf{P}})$
9:          Calculate equation loss: $L_{eq} = \text{residual}(\hat{\mathbf{u}})$
10:         Calculate total loss: $L_{\text{total}} = c_1 \cdot L_u + c_2 \cdot L_s + c_3 \cdot L_{eq}$
11:         Backpropagate total loss and update FNO model
12:     **end for**
13:     Evaluate on validation set
14:     Record training and validation loss
15: **end for**

$\triangleright$ $c_1$, $c_2$, and $c_3$ are learnable coefficients Liebel & Körner (2018)

---

## B    DIFFERENTIAL EQUATION DETAILS

In this section, the differential equations investigated in the work are detailed. This includes their mathematical formulations, initial conditions, parameter setups.

**ODE1: Composite Harmonic Oscillator** We chose this ODE because it has an analytical solution, allowing us to validate our differential equation solvers and sensitivity computations. The ODE is defined as:

$$\frac{du}{dt} = \alpha \sin(\alpha \pi t) + \beta \cos(\beta \pi t) \tag{7}$$

with the initial condition $u(0) = \sin(\gamma \pi)$. This oscillator's behavior is modulated by the parameters $\alpha$, $\beta$, and $\gamma$, affecting the frequency and amplitude of oscillations within the temporal domain $t \in [0, 1]$. The analytical solution for u(t) is:

$$u(t) = -\frac{1}{\pi} \cos(\alpha \pi t) + \frac{1}{\pi} \sin(\beta \pi t) + \sin(\gamma \pi) + \frac{1}{\pi} \tag{8}$$

The sensitivities of u with respect to each parameter are:

$$\frac{\partial u}{\partial \alpha} = t \sin(\alpha \pi t) \tag{9}$$

$$\frac{\partial u}{\partial \beta} = t \cos(\beta \pi t) \tag{10}$$

$$\frac{\partial u}{\partial \gamma} = \pi \cos(\gamma \pi) \tag{11}$$

These analytical solutions provide a benchmark against which we can compare the accuracy of our numerical solvers and sensitivity computations.

**ODE2: Duffing Oscillator Equation**

$$\ddot{x} + \delta \dot{x} + \alpha t + \beta t^3 = \gamma \cos(\omega t), \tag{12}$$

with initial conditions $x(0) = \epsilon, \dot{x}(0) = \zeta$. This equation describes a non-linear oscillator where damping $\delta$, stiffness $\alpha$, non-linear stiffness $\beta$, driving amplitude $\gamma$, and frequency $\omega$ play crucial roles.

**PDE1: Generalized Nonlinear Damped Wave Equation**

$$\frac{\partial^2 u}{\partial t^2} = c^2 \frac{\partial^2 u}{\partial x^2} + \alpha \frac{\partial u}{\partial t} + \beta u + \gamma \sin(\omega u), \tag{13}$$

with initial conditions $u(x,0) = u_0$, $\frac{\partial u}{\partial t}(x,0) = u_0'$. This PDE extends over a spatial domain $x \in [0, 1]$ and temporal domain $t \in [0, 1]$, exploring wave propagation influenced by damping $\alpha$, stiffness $\beta$, and external forcing $\gamma$ and $\omega$.

**PDE2: Forced Burgers' Equation**

$$\frac{1}{\pi} \frac{\partial u}{\partial t} + \alpha u \frac{\partial u}{\partial x} = \gamma \frac{\partial^2 u}{\partial x^2} + \delta \sin(\omega t), \tag{14}$$

This equation is set within a spatial domain $x \in [0, 1.0]$ and a temporal domain $t \in [0, \pi]$. It models fluid dynamics phenomena such as velocity $u(x, t)$, incorporating effects of advection $\alpha$, viscosity $\gamma$, and external periodic forcing characterized by amplitude $\delta$ and frequency $\omega$. The initial state of the system is defined as follows:

$$u(x,0) = u_0(x) = \left( e^{-\frac{(x-x_0)^2}{2\sigma^2}} + \sin(0.5\pi x) \right), \tag{15}$$

where the Gaussian pulse is centered at $x_0 = 0.5$ with a width $\sigma = 0.3$, combined with a sinusoidal component. The model employs periodic boundary conditions, ensuring that $u(0, t) = u(1.0, t)$

throughout the simulation, facilitating the study of continuous and cyclic phenomena in a finite spatial interval.

**PDE3: Stream Function-Vorticity Formulation of the Navier-Stokes Equations**

$$\frac{\partial \omega}{\partial t} + \psi_y \frac{\partial \omega}{\partial x} - \psi_x \frac{\partial \omega}{\partial y} = \frac{1}{Re}\left(\frac{\partial^2 \omega}{\partial x^2} + \frac{\partial^2 \omega}{\partial y^2}\right), \tag{16}$$

$$\frac{\partial^2 \psi}{\partial x^2} + \frac{\partial^2 \psi}{\partial y^2} = -\omega, \tag{17}$$

with initial condition $\omega(x, y, 0) = f(x, y; \alpha, \beta)$ where:

$$f(x, y; \alpha, \beta) = \sin(\alpha x)\cos(\beta y) + \cos(\alpha y)\sin(\beta x) + \sin(\alpha x + \beta y)\cos(\alpha y - \beta x), \tag{18}$$

covering the spatial domain $x, y \in [0, 1]$ and temporal domain $t in [0, 3]$. The Reynolds Number $Re$ for the Navier-Stokes equations is set to 1000 to simulate realistic fluid dynamics. This equation captures the dynamics of fluid flow, with initial vorticity distribution determined by parameters $\alpha$ and $\beta$.

**PDE4: Allen-Cahn equation**

$$\frac{\partial u}{\partial t} = \epsilon \frac{\partial^2 u}{\partial x^2} + \alpha u - \beta u^3, \tag{19}$$

with initial condition $u(x, 0) = c\tanh(\omega x)$ and periodic boundary conditions. This PDE, known for exhibiting rich bifurcation behavior, explores phase transition phenomena influenced by diffusion coefficient $\epsilon$, linear term $\alpha$, cubic term $\beta$, and initial condition parameters $c$ and $\omega$. The equation's solutions can undergo sharp qualitative changes with small parameter variations, making it an excellent test case for sensitivity analysis.

The parameter ranges used in our simulations are detailed in Table B.6. Parameters for these simu-

Table B.6: Parameter values for different ODE and PDE cases.

| Case | $c$ | $\alpha$ | $\beta$ | $\gamma$ | $\delta$ | $\omega$ | $\epsilon$ | $\zeta$ | $M$ |
|---|---|---|---|---|---|---|---|---|---|
| ODE 1 | - | [1, 3] | [1, 3] | [0, 1] | - | - | - | - | 10 |
| ODE 2 | - | [0.02, 0.06] | [0.01, 0.03] | [20, 60] | [0.5, 1.5] | [0.2, 0.6] | [0.0, 0.2] | [0.0, 0.2] | 10 |
| PDE 1 | [0.0, 0.25] | [0.0, 0.1] | [0.0, 0.25] | [0.0, 0.25] | - | [0.0, 0.25] | - | - | 5 |
| PDE 2 | - | [0.1, 1.0] | - | [0.025, 0.25] | [0.1, 0.5] | [0.01, 0.1] | - | - | 5 |
| PDE 3 | - | $[\pi, 5\pi]$ | $[\pi, 5\pi]$ | - | - | - | - | - | 1 |
| PDE 4 | [0.1,0.9] | [0.01,1.0] | [0.01,1.0] | - | - | [5.0, 10.0] | [0.01,1.0] | - | 5 |

lations were randomly generated using a uniform distribution. The uniform distribution is denoted by $\mathcal{U}(a, b)$, where $a$ is the lower bound and $b$ is the upper bound of the distribution. This means that any value within the range $[a, b]$ has an equal probability of being selected. The uniform distribution was chosen to ensure a balanced representation of parameter values across the entire specified range, without favoring any particular subset of values. This approach allows for a comprehensive exploration of the parameter space, providing a robust test of our models across a wide range of potential input conditions.

## C  FNOS HYPERPARAMETERS

Table C.7 presents the hyperparameters used for training the FNO models for each case study. In the network architecture, "Mode" refers to the Fourier modes used in the neural network's layers for each dimension (t, x, and y), while "Width" denotes the number of channels or features in the hidden layers of the neural network. The learning rate and number of epochs used for training are also provided for each case. Additionally, Table C.8 compares training times per epoch for different model configurations and batch sizes.

Table C.7: Hyperparameters for FNOs

| Case | Mode for t | Mode for x | Mode for y | Width | Number of Fourier Layers | Learning Rate | Number of Epochs | Number of Learnable Parameters |
|------|-----------|-----------|-----------|-------|-------------------------|---------------|-----------------|-------------------------------|
| ODE 1 | 8 | 8 | - | 20 | 4 | 0.001 | 500 | 17921 |
| ODE 2 | 8 | 8 | - | 20 | 4 | 0.001 | 500 | 17921 |
| PDE 1 | 8 | 8 | - | 20 | 4 | 0.001 | 500 | 107897 |
| PDE 2 | 8 | 8 | 8 | 20 | 4 | 0.001 | 500 | 107897 |
| PDE 3 | - | 8 | 8 | 20 | 4 | 0.001 | 500 | 209397 |
| PDE 4 | 8 | 8 | 8 | 20 | 4 | 0.001 | 500 | 107897 |

Table C.8: Comparison of model configurations and training time

| Case | Batch size | Number of physical parameters (p) | Number of training samples | Average training time per epoch (s) | | | |
|------|-----------|-----------------------------------|----------------------------|------|--------|----------|-------------|
| | | | | FNO | SC-FNO | FNO-PINN | SC-FNO-PINN |
| ODE1 | 16 | 3 | 2000 | 1.10 | 1.94 | 1.53 | 2.46 |
| ODE2 | 16 | 7 | 2000 | 1.58 | 2.13 | 1.76 | 2.86 |
| PDE1 | 4 | 5 | 2000 | 35.24 | 53.32 | 52.13 | 82.13 |
| PDE2 | 4 | 4 | 2000 | 32.66 | 44.92 | 39.11 | 73.06 |
| PDE2 (Zoned) | 1 | 82 | 100 | 5.37 | 7.23 | - | - |
| PDE2 (Zoned) | 1 | 82 | 500 | 8.09 | 11.23 | - | - |
| PDE3 | 4 | 2 | 1000 | 47.16 | 109.43 | - | - |
| PDE4 | 1 | 5 | 100 | 11.54 | 19.12 | - | - |

# D ADDITIONAL RESULTS

## D.1 PERFORMANCE COMPARISON OF NEURAL OPERATORS

This appendix section presents a comprehensive comparison of the performance of various neural operators, both in their original form and with our proposed sensitivity-constrained framework. We evaluate these operators on four different systems, PDE1 and PDE2. The following tables provide quantitative results for Fourier Neural Operators (FNO), Wavelet Neural Operators (WNO), Multiwavelet Neural Operators (MWNO), and DeepONets, along with their sensitivity-constrained counterparts.

Table D.9: $R^2$ for PDE1 with $2 \times 10^3$ training samples.

| Value | FNO | SC-FNO | WNO | SC-WNO | MWNO | SC-MWNO | DeepONet | SC-DeepONet |
|---|---|---|---|---|---|---|---|---|
| $u(t)$ | **0.986** | 0.983 | 0.981 | **0.989** | 0.978 | 0.952 | 0.974 | 0.954 |
| $\frac{\partial u}{\partial c}$ | 0.723 | 0.924 | 0.661 | 0.919 | 0.596 | **0.939** | 0.117 | 0.508 |
| $\frac{\partial u}{\partial \alpha}$ | 0.741 | **0.925** | 0.402 | 0.923 | 0.090 | 0.916 | 0.110 | 0.243 |
| $\frac{\partial u}{\partial \beta}$ | 0.763 | **0.930** | 0.921 | 0.914 | 0.863 | 0.915 | 0.778 | 0.827 |
| $\frac{\partial u}{\partial \gamma}$ | 0.772 | **0.931** | 0.572 | 0.834 | 0.531 | 0.659 | 0.483 | 0.572 |
| $\frac{\partial u}{\partial \omega}$ | 0.781 | **0.932** | 0.614 | 0.823 | 0.558 | 0.684 | 0.497 | 0.531 |

Table D.10: $R^2$ for PDE2 with $2 \times 10^3$ training samples.

| Value | FNO | SC-FNO | WNO | SC-WNO | MWNO | SC-MWNO | DeepONet | SC-DeepONet |
|---|---|---|---|---|---|---|---|---|
| $u(t)$ | 0.997 | **0.997** | 0.981 | 0.979 | 0.989 | 0.971 | 0.990 | 0.985 |
| $\frac{\partial u}{\partial \alpha}$ | 0.206 | **0.987** | 0.422 | 0.635 | 0.558 | 0.821 | 0.344 | 0.882 |
| $\frac{\partial u}{\partial \gamma}$ | 0.423 | **0.986** | 0.432 | 0.562 | 0.387 | 0.905 | 0.511 | 0.873 |
| $\frac{\partial u}{\partial \delta}$ | 0.821 | 0.912 | 0.865 | 0.894 | 0.501 | 0.932 | 0.445 | **0.958** |
| $\frac{\partial u}{\partial \omega}$ | 0.321 | **0.982** | 0.425 | 0.542 | 0.214 | 0.952 | 0.363 | 0.806 |

## D.2 ADDITIONAL RESULTS FOR THE INVERSION EXPERIMENTS

Figures D.8, D.9 and D.10 showcase parameter inversion results for parameters for PDE1, PDE2, and PDE3, respectively. SC-FNO works significantly better than either FNO or FNO-PINN. Table D.11 shows the comparison between various neural operators and their sensitivity-constrained (SC-) versions in simultaneous parameter inversion. Note the uniform pattern that the sensitivity-constrained versions have much higher inversion accuracy. Furthermore, the differences between different neural operators are smaller than the difference between the versions with and without the sensitivity constraint.

Table D.11: Multi-parameter inversion accuracy for PDEs 1 and 2 with and without gradient supervision.

(a) PDE1

| | Fourier Neural Operators | | | | Other Neural Operators | | | | | |
| | FNO | | SC-FNO | | Without gradient supervision. | | | With gradient supervision. | | |
| Parameter | R² | Relative L² | R² | Relative L² | WNO | MWNO | DeepONet | SC-WNO | SC-MWNO | SC-DeepONet |
|---|---|---|---|---|---|---|---|---|---|---|
| c | 0.657 | 0.212 | 0.987 | 0.035 | 0.636 | 0.614 | 0.538 | 0.984 | 0.981 | 0.977 |
| $\alpha$ | 0.642 | 0.222 | 0.986 | 0.036 | 0.621 | 0.598 | 0.519 | 0.984 | 0.980 | 0.977 |
| $\beta$ | 0.753 | 0.183 | 0.974 | 0.042 | 0.738 | 0.723 | 0.668 | 0.969 | 0.963 | 0.956 |
| $\gamma$ | 0.804 | 0.165 | 0.985 | 0.037 | 0.792 | 0.780 | 0.736 | 0.982 | 0.978 | 0.975 |
| $\omega$ | 0.750 | 0.186 | 0.916 | 0.075 | 0.735 | 0.719 | 0.663 | 0.901 | 0.879 | 0.859 |

(b) PDE2

| | Fourier Neural Operators | | | | Other Neural Operators | | | | | |
| | FNO | | SC-FNO | | Without Grad. Sup. | | | With Grad. Sup. | | |
| Parameter | R² | Relative L² | R² | Relative L² | WNO | MWNO | DeepONet | SC-WNO | SC-MWNO | SC-DeepONet |
|---|---|---|---|---|---|---|---|---|---|---|
| $\alpha$ | 0.954 | 0.078 | 0.982 | 0.045 | 0.951 | 0.949 | 0.938 | 0.979 | 0.974 | 0.970 |
| $\gamma$ | 0.925 | 0.082 | 0.967 | 0.022 | 0.921 | 0.916 | 0.899 | 0.961 | 0.953 | 0.945 |
| $\delta$ | 0.842 | 0.145 | 0.970 | 0.051 | 0.832 | 0.822 | 0.787 | 0.964 | 0.956 | 0.949 |
| $\omega$ | 0.895 | 0.118 | 0.986 | 0.042 | 0.889 | 0.838 | 0.859 | 0.983 | 0.973 | 0.976 |

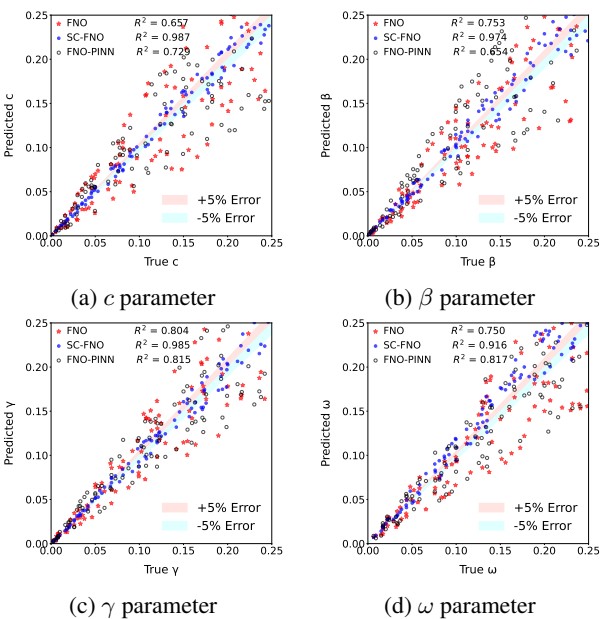

Figure D.8: Individual parameter results for PDE1 in simultaneous multi-parameter recovery, comparing FNO and SC-FNO performance. ($\alpha$ has been presented in Figure 1b)

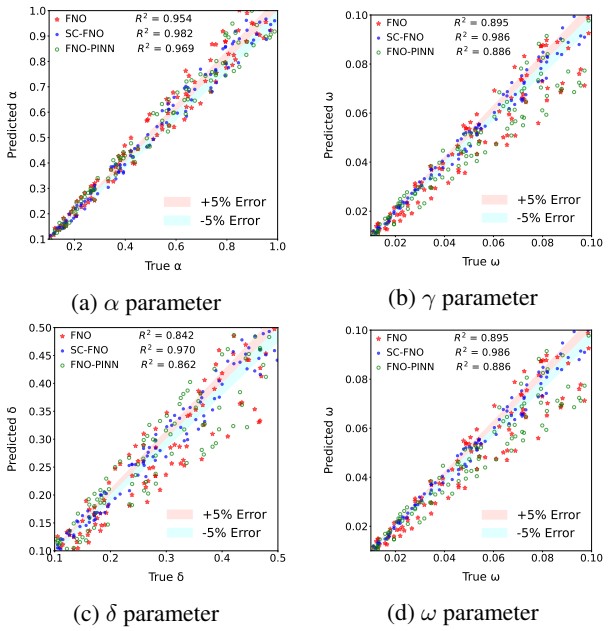

Figure D.9: Individual parameter results for PDE2 in simultaneous multi-parameter recovery, comparing FNO and SC-FNO performance.

## D.3 VERIFICATION OF DIFFERENTIABLE AND FINITE DIFFERENCE SOLVERS

Table D.13 presents a comparison of $R^2$ values for both the solution path u(t) and the gradients of u with respect to different parameters obtained from both differentiable solver (AD) and 4th order finite difference solver (FD). The table includes runtime measurements for generating $2 \times 10^3$ samples using both solvers, providing insight into their computational efficiency. These measurements were obtained using a machine equipped with a V100 GPU and four Intel Xeon processors, offering a standardized comparison of computational cost (The Wall Clock Times) between the two solvers.

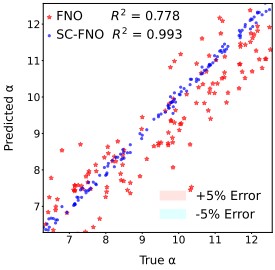

Figure D.10: Inversion of the parameter $\alpha$ in PDE3 using FNO and SC-FNO models.

Table D.12: Computation time for preparing datasets using AD solver with and without Jacobian

| Case | Number of input parameter (P) | Computation time (s) | |
|---|---|---|---|
| | | With jacobian | Without jacobian |
| PDE1 | 5 | 1.852 | 0.932 |
| PDE2 | 4 | 1.387 | 0.796 |
| PDE3 | 2 | 6.205 | 2.762 |

Table D.13: Error metrics and runtime comparison for training data preparation with AD and FD solvers on ODE1

| | FD solver | | AD solver | |
|---|---|---|---|---|
| Value | $R^2$ | Runtime | $R^2$ | Runtime |
| $u$ | 0.9872 | | **0.9981** | |
| $\frac{\partial u}{\partial \alpha}$ | 0.9712 | 1907.32 s | **0.9906** | 252.54 s |
| $\frac{\partial u}{\partial \beta}$ | 0.9885 | | **0.9989** | |
| $\frac{\partial u}{\partial \gamma}$ | 0.9618 | | **0.9890** | |

## D.4 SURROGATE MODEL QUALITY FOR THE ORDINARY DIFFERENTIAL EQUATIONS (ODES).

Table D.14: Test $R^2$ values and Relative L² for the solution paths $u$ and sensitivities ($\frac{\partial u}{\partial p}$) of the surrogate models for ODE1 and ODE2 with $2 \times 10^3$ training samples. The train and test parameters are in the same range. ODEs are simpler to capture by surrogate models than PDEs.

(a) ODE1

| Value | Metric | FNO | SC-FNO | SC-FNO-PINN | FNO-PINN |
|---|---|---|---|---|---|
| $u(t)$ | R² | **0.996** | 0.991 | 0.994 | 0.991 |
| | Relative L² | **0.004** | 0.009 | 0.006 | 0.009 |
| $\frac{\partial u}{\partial \alpha}$ | R² | 0.327 | **0.994** | 0.992 | 0.318 |
| | Relative L² | 0.673 | **0.006** | 0.008 | 0.682 |
| $\frac{\partial u}{\partial \beta}$ | R² | 0.415 | **0.995** | **0.995** | 0.462 |
| | Relative L² | 0.585 | **0.005** | **0.005** | 0.538 |
| $\frac{\partial u}{\partial \gamma}$ | R² | 0.028 | **0.998** | **0.998** | 0.131 |
| | Relative L² | 0.972 | **0.002** | **0.002** | 0.869 |

(b) ODE2

| Value | Metric | FNO | SC-FNO | SC-FNO-PINN | FNO-PINN |
|---|---|---|---|---|---|
| $u(t)$ | R² | **0.998** | 0.995 | 0.997 | 0.997 |
| | Relative L² | **0.002** | 0.005 | 0.003 | 0.003 |
| $\frac{\partial u}{\partial \alpha}$ | R² | 0.162 | 0.997 | **0.998** | 0.152 |
| | Relative L² | 0.838 | 0.003 | **0.002** | 0.848 |
| $\frac{\partial u}{\partial \delta}$ | R² | 0.956 | 0.997 | **0.998** | 0.951 |
| | Relative L² | 0.044 | 0.003 | **0.002** | 0.049 |
| $\frac{\partial u}{\partial \zeta}$ | R² | 0.135 | 0.996 | **0.997** | 0.147 |
| | Relative L² | 0.865 | 0.004 | **0.003** | 0.853 |

