# OpenReview forum: "Sensitivity-Constrained Fourier Neural Operators for Forward and Inverse Problems in Parametric Differential Equations"
_ICLR.cc/2025/Conference — ICLR 2025 Poster_

### Official Review · Reviewer_WFXg · 2024-10-20

**Soundness:** 2
**Presentation:** 2
**Contribution:** 3
**Rating:** 6
**Confidence:** 4

**Summary:**

This paper proposes a sensitivity-constrained loss in the original FNO model, making it suitable for ODE/PDE inversion problems. Definitely this loss can make the prediction more stable with respect to the physical system parameters. This loss works with a wide range of neural operators and other supervising sensitivity tasks. The paper is well-written. The method is effective compared to the original FNO and FNO-PINN. The experiments on five ODE/PDE problems are impressive.

**Strengths:**

1.The paper focus on the sensitivity prediction to physical ODE/PDE systems, which is important.

2.The paper’s contribution is clear, and the paper is easy to follow.

3.The experiments are diverse.

4.The paper’s improvements help enhancing the reliability and applicability of neural operators in complex physical systems modeling and engineering analysis.

**Weaknesses:**

1.The idea of using gradient loss to guarantee sensitivity is not new, such as the gradient PINN and others.

2.The sensitivity loss is supervised, which means additional labeled data {\partial u}/{\partial p} are needed compared to original FNO. On the other hand , if we have additional labeled data, it is directly to add the supervised loss in the total loss, which is SC-FNO exactly, so I think the novelty is rather limited. Although the {\partial u}/{\partial p}  can also be approximately computed from the original data, the numerical error influence is not analyzed. On the contrary, the PINN loss is unsupervised.

3.The SC-FNO loss is hybrid, the training process should be reflected in the main text or appendix. SC-FNO should be hard to train compared to FNO, but the author didn’t mention the difference.

4.Since the PINN-FNO also add the PINN loss as the regularizer to guarantee the prediction of physical parameters, I wonder why it behaves bad to SC-FNO. Maybe the training difficulty is different. The author use a learnable coefficients combination of the hybrid loss Liebel & Körner (2018) in Algorithm 3, but FNO-PINN may be trained well for adaptive hybrid loss as in Wang S, Wang H, Perdikaris P. Improved architectures and training algorithms for deep operator networks[J]. Journal of Scientific Computing, 2022, 92(2): 35.

5. SC-FNO architecture,  as well as the Algorithm, should be exhibited in the main text,  not the appendix.

**Questions:**

See in the weaknesses 2-4.

Why use R^2 as metric, why not the relatively L^2 norm as in the original FNO paper？The metric should be comparable to existing works.

---

> ### Author Response · Authors · 2024-11-19
>
> Thank you for your comments which have provided an opportunity to clarify key aspects of our work and the critical difference.
>
> ----
> ## Part 1
>
> **Weakness 1. Novelty and Fundamental Differences from PINNs:** **SC-FNO uniquely adds time-integrated parameter sensitivities (∂u/∂p) as a supervisory signal** (Lines 368-372). PINNs do not have access to this information, as usual PDEs do not contain ∂u/∂p, leaving it unconstrained.
> Procedurally, SC-FNO and PINNs differ entirely: SC-FNO utilizes a forward numerical model with differentiable solvers (or finite difference, or adjoint) to prepare data to supervise ∂u/∂p, whereas PINNs rely on collocation points for equation-based loss optimization to supervise (∂u/∂x, ∂u/∂t), but not ∂u/∂p.
>
> The experimental benefits of ∂u/∂p are clear:
>   - SC-FNO consistently achieves R² > 0.94 for all parameters in PDE1, surpassing FNO and FNO-PINN which remain below 0.82.
>   - In parameter inversion tasks, SC-FNO reaches near-perfect recovery (R² = 0.998), significantly outperforming FNO (R² = 0.905).
>   - For complex systems like Navier-Stokes equations, SC-FNO enhances solution accuracy (R² from 0.883 to 0.994) and sensitivity estimation (R² from 0.246 to 0.997).
>
> Given these profound differences in theory, procedure, and performance, the critique of lacking novelty is surprising. **Is it fair to dismiss the differentiated outcomes in Figure 1? Does using gradient terms in the loss truly negate novelty?**
>
> ---
>
> **Weakness 2. Data Efficiency and Computational Considerations:** Our experiments confirm that leveraging sensitivity information via our SC-FNO model enhances both efficiency and robustness. Specifically:
>
> - **Computational Efficiency:** Gradient information, a byproduct of our differentiable solver, increases computation time only moderately. Memory overhead remains low (722MB for SC-FNO vs. 764MB for FNO). Dataset preparation time was in the original Table D.9. For PDE2 (with 1300 training samples), training time per epoch increased only from 21 to 28 seconds. Also, see the new example below for training time and L2 error. We will add training time info to all Tables.
>
> - **Reduced Data Requirements:** SC-FNO matches the accuracy of FNO with four times fewer data points (500 vs. 2000, original Figure 6). Under limited data conditions (500 samples), SC-FNO (R²>0.9) is significantly better than FNO’s (R²=0.8).
>
> - **Robust Generalization:** SC-FNO sustains accuracy with parameter perturbations up to 40% beyond the training range and excels in parameter inversion tasks, boosting R² from 0.642 to 0.986 for PDE1 (original Figure 1).
>
> In summary, SC-FNO's sensitivity supervision not only minimizes data needs but also ensures greater generalization, more than offsetting the slight increase in computational demands.
>
> We also refined our experiment with PDE2 (Burger's equation), segmenting the spatial domain into 40 zones. Each zone had specific advection (α) and forcing amplitude (δ), alongside global parameters viscosity (γ) and frequency (ω), totaling 82 parameters (2*M+2). The results are:
>
> |Number of training samples | N=500 | | N=100 | |
> |---------------------------------|---------------------|--------------|---------------------|--------------|
> | Model | SC-FNO | FNO | SC-FNO | FNO |
> | *Average training time per epoch (s)* | 11.23 | 8.09 | 7.23 | 5.37 |
> | *State Value Metrics* | | | | |
> | Relative L2 | 0.0073 | 0.0282 | 0.0087 | 0.0387 |
> | Maximum Absolute Error | 0.0543 | 0.3056 | 0.0568 | 0.2831 |
> | *Mean Jacobian Metrics* | | | | |
> | Relative L2 | 0.1770 | 1.9627 | 0.2134 | 2.4623 |
> | Maximum Absolute Error | 0.0558 | 0.2008 | 0.0653 | 0.2208 |
>
> These new results (to be included in revision) show that, with higher input dimensions, the default FNO had a much larger error for the state variable while FNO stayed highly accurate.
>
> ---
>
>
> **Weakness 3. “SC-FNO Should Be Harder to Train”:**
>
> The FNO's direct work in Fourier space actually helps with SC-FNO training, not hinders it. Our experiments show this: training remains stable with learnable loss coefficients and adds minimal overhead (only 10% computation time, 5.8% memory increase from 722MB to 764MB). Instead of making training harder, the sensitivity information actually helps guide the model to better solutions - that's why we see such improved performance across different equations. The complete process detailed in Algorithms 1-3 shows how this straightforward approach brings major benefits without significant complexity.

---

> > ### Author Response · Authors · 2024-11-19
> >
> > ## Part 2
> >
> > ---
> >
> > **Weakness 4. PINN + FNO vs. PINN + DeepOnet:**
> > The paper you referenced (Wang et al., 2022) specifically demonstrates successful PINN integration with DeepONet architecture, not FNO. The contrasting performance may stem from fundamental architectural differences in how these neural operators handle derivatives. DeepONet directly operates in function spaces through its branch and trunk networks, making PINN's differential operators naturally compatible. **In contrast, FNO transforms data to Fourier space where derivatives become multiplication operations, making spatial/temporal derivative supervision less effective.**
> > This architectural distinction explains why PINN regularization works better with DeepONet than FNO, despite both being neural operators. While both approaches map between function spaces, their internal operations are fundamentally different. This core difference in handling derivatives suggests why SC-FNO's direct parameter sensitivity supervision proves more effective for FNO-based architectures.
> >
> > ---
> >
> > **Question 1. Use of R²:**
> > We will include additional error metrics such as relative L2 error in the revised manuscript.
> > Below is a sample calculation of additional error metrics for PDE2 (Burger's equation):
> > | Metric          | SC-FNO | FNO  |
> > |-----------------|-------------|-----------|
> > | **State Value Metrics** |             |           |
> > | RMSE            | 0.00221     | 0.00411   |
> > | L2              | 0.85780     | 1.59028   |
> > | Rel L2          | 0.00161     | 0.00299   |
> > | Max L1          | 0.01363     | 0.05133   |
> > | **Mean Jacobian Metrics** |             |           |
> > | RMSE            | 0.01376     | 0.24827   |
> > | L2              | 5.32954     | 96.15426  |
> > | Rel L2          | 0.01349     | 0.20916   |
> > | Max L1          | 0.20029     | 1.53333   |
> >
> > ---

---

> > > ### Comment · Reviewer_WFXg · 2024-11-20
> > >
> > > Thanks for the detailed reply and additional experiments.
> > >
> > > I agree with your explanation on the training and the experiments, although the improvements are mainly on ∂u/∂p(rel.l2 from 0.21 to 0.013) rather than on u itself(from 0.003 to 0.0016).
> > >
> > > My major concern (maybe personally) about the novelty is that, SC-FNO need additional supervised data ∂u/∂p compared to FNO, while FNO-PINN do not need additional supervised data but the known equation. From FNO to FNO-PINN is innovative since we can generalize well without additional labelled data, but from FNO to SC-FNO we need additional labelled data to achieve better performance. The data is in a typical supervised form and can be added directly to the loss, so SC-FNO is very straightforward. In this case I think SC-FNO has less contribution to the neural network architecture or the loss design.

---

> > > > ### Author Response · Authors · 2024-11-21
> > > >
> > > > Thank you for the interesting observation.
> > > >
> > > > Another way to view this conversation about novelty is the following. The SC-FNO term does not fit into the standard loss framework because the loss term supervises the gradient of the network (whereas the standard loss framework supervises the output of the network). In a sense, SC-FNO is a novel variant of PINN. Roughly, PINNs are loss terms to enforce **f(network derivatives)=0**, while SC-FNO adds a loss term to enforce **f(network derivatives)=data - dependent**. Aside from this, the preparation of new information in the data is also an important novel contribution that should not be overlooked.

---

> > > > > ### Comment · Reviewer_WFXg · 2024-11-24
> > > > >
> > > > > Thank you for your careful reply. Now I know that the SC-FNO's additional supervised loss isn't a 'typical' supervised loss. Identically it has the same data requirement as FNO (input: function a(x), parameter p, coordinates (t,x); output: u(x)).
> > > > > The additional 'supervised' data ∂u/∂p can be computed from the original output data numerically, and the prediction can be computed by automatic differentiation. I agree with the author that "SC-FNO is a novel variant of PINN".
> > > > >
> > > > > All of my concerns have been clarified. I'd like to increase the score to a positive score and hope the authors clarify well in a revised version of the paper.

---

> > > > > > ### Author Response · Authors · 2024-11-24
> > > > > >
> > > > > > Thank you for your feedback. We believe your insights will greatly enhance the quality of our work.
> > > > > >
> > > > > > We look forward to addressing your points clearly in our revised manuscript.

---

> ### Author Response · Authors · 2024-11-26
>
> Dear reviewer,
>
> We greatly appreciate your thorough review, which has helped improve the quality of our manuscript. We have carefully considered all your comments and have updated our manuscript accordingly to address your concerns.
>
> We would be most grateful for your input if any additional revisions are needed during the remaining time of the discussion period.

---

### Official Review · Reviewer_KLdr · 2024-11-03

**Soundness:** 3
**Presentation:** 3
**Contribution:** 3
**Rating:** 6
**Confidence:** 4

**Summary:**

This paper introduces a sensitivity loss regularizer for neural operators, specifically applying it to the Fourier Neural Operator (FNO) framework, resulting in Sensitivity-Constrained Fourier Neural Operators (SC-FNO). SC-FNO improves accuracy in solution paths, inverse problem-solving, and sensitivity calculations, even under sparse data and concept drift. It outperforms both standard FNO and FNO with PINN regularization in parameter inversion tasks, achieving high accuracy without large computational or memory costs. These enhancements extend the applicability and reliability of neural operators in modeling complex physical systems.

**Strengths:**

1. The paper's proposed Sensitivity-Constrained approach is applicable to various neural operators, offering high research value.
2. The paper is clearly written, with no redundant mathematical derivations, and offers excellent readability.
3. In modeling PDEs, this approach not only enables accurate approximation of the target variable but also effectively models its derivatives.

**Weaknesses:**

1. There is no enough discussion on the memory overhead introduced by AD (Automatic Differentiation) and FD (Finite Differences).
2. The experiments are limited. It is not sure whether this method remains effective in complicated and high-resolution PDE scenarios.

**Questions:**

In above part.

---

> ### Author Response · Authors · 2024-11-20
>
> We sincerely appreciate the thoughtful and detailed feedback provided in your review. Your comments have identified key areas where further clarification could significantly strengthen our manuscript.
>
> ---
> ## Part 1
>
> **Question 1 (Discussion on Memory Overhead in AD and FD):**
> Our discussion in Section 3.4 primarily focuses on demonstrating the feasibility of computing Jacobians using both traditional finite difference methods and automatic differentiation approaches.
> The experimental results show that both methods are practical, with AD being more efficient. As detailed in Table D.9, generating 2000 samples with gradients takes 252.54 seconds using AD versus 1907.32 seconds using finite differences, while achieving comparable accuracy (R² > 0.96). The memory overhead remains modest - training FNO on PDE1 used 722MB while SC-FNO required 764MB, representing only a 5.8% increase. We would be happy to provide a more detailed profiling of memory usage in the revised manuscript for other cases, but generally, the requirements are quite modest.
>
> ---

---

> > ### Author Response · Authors · 2024-11-20
> >
> > ## Part 2
> >
> > **Question 2 (Assessing Effectiveness in Complex PDE Scenarios: Additional Equations):**
> > In our manuscript, we investigated our framework across a variety of differential equations, from simple ODEs to complex PDEs like Navier-Stokes equations. To further demonstrate SC-FNO's capabilities, we conducted two new challenging experiments:
> >
> > * First, we tested Burger's equation (PDE2) with zoned parameters - dividing the spatial domain into M=40 segments with different advection α and forcing amplitude δ in each zone, along with global parameters viscosity γ and forcing frequency ω (total 2M+2=82 parameters). The results demonstrate SC-FNO's clear advantages over FNO (Table below):
> >
> >
> > |Number  of training samples | N=500 | | N=100 | |
> > |---------------------------------|---------------------|--------------|---------------------|--------------|
> > | Model | SC-FNO | FNO | SC-FNO | FNO |
> > | **Average training time per epoch (s)** | 11.23 | 8.09 | 7.23 | 5.37 |
> > | **State Value Metrics** | | | | |
> > | RMSE | 0.01006 | 0.03894 | 0.01199 | 0.05354 |
> > | Relative L2 | 0.00729 | 0.02822 | 0.00867 | 0.03873 |
> > | Maximum Absolute Error | 0.05425 | 0.30564 | 0.05679 | 0.28312 |
> > | **Mean Jacobian Metrics** | | | | |
> > | RMSE | 0.00508 | 0.05181 | 0.00660 | 0.06076 |
> > | Relative L2 | 0.17698 | 1.96270 | 0.21342 | 2.46226 |
> > | Maximum Absolute Error | 0.05576 | 0.20075 | 0.06531 | 0.22079 |
> >
> >
> >
> > * Second, we examined Navier-Stokes equations with specifics introduced in the manuscripts and initial condition *w₀(x)* is generated according to *w₀ ~ μ* where *μ = N(0, 7^(3/2) (-∆+49I)^(-2.5))* with periodic boundary conditions (according Li, Zongyi, et al. "Fourier neural operator for parametric partial differential equations." *arXiv preprint arXiv:2010.08895* (2020).) and also varied Reynolds number in range of [250 to 1000] corresponding to kinematic viscosity [0.004 to 0.001] and constrain SC-FNO with jacobian of solution path (vorticity) with respect to the Reynolds number with different training sample points. Results show SC-FNO's performance (Table below):
> >
> > | Number  of training samples | | N=200 | | N=100 | |
> > |---------------------------------------|-------------|---------------|--------------|---------------|-------------|
> > | Model | **FNO** | **SC-FNO** | **FNO** | **SC-FNO** |
> > | **Average training time per epoch (s)** | 4.46 | 11.27 | 2.51 | 6.13 |
> > | **State Value Metrics** | | | | |
> > | vorticity R2 Score | 0.844 | 0.996 | 0.804 | 0.986 |
> > | vorticity L2 Mean Error (RMSE) | 1.36e-01 | 2.28e-02 | 1.53e-01 | 4.12e-02 |
> > | vorticity L1 Relative Error | 0.389 | 0.066 | 0.452 | 0.120 |
> > | vorticity L2 Relative Error | 0.390 | 0.066 | 0.442 | 0.119 |
> > | vorticity Max Absolute Error | 0.747 | 0.153 | 0.728 | 0.216 |
> > | **Mean Jacobian Metrics** | | | | |
> > | Jacobian R2 Score | 0.170 | 0.996 | 0.181 | 0.990 |
> > | Jacobian Mean L2 Error (RMSE) | 2.32e-03 | 1.65e-04 | 2.31e-03 | 2.53e-04 |
> > | Jacobian L1 Relative Error | 1.510 | 0.105 | 1.416 | 0.157 |
> > | Jacobian L2 Relative Error | 0.877 | 0.062 | 0.872 | 0.096 |
> > | Jacobian Max Absolute Error | 0.122 | 0.005 | 0.259 | 0.012 |
> >
> > These additional results further confirm our method's effectiveness in both high-dimensional and complex physical systems.
> >
> > We nonetheless apologize for the oversight of not providing computational time and relative L2 error information. In fact, when you look at the relative L2 error in the last two tables, SC-FNO’s errors even for the state variables are several times smaller than that of FNO, with only a modest increase in training time in this case. It is data-efficient --- N=100 gives us a much better model than N=500 for the default FNO. It extrapolates much more robustly (original Figure 4). Furthermore, the advantages over FNO grow much more significantly as the dimension grows -- while its training time understandably will increase, the data demand for FNO will increase exponentially. We mention these factors for the reviewers’ kind consideration. All these results point to an enormous elevation of neural operators’ performance ceiling.

---

> ### Author Response · Authors · 2024-11-26
>
> Dear reviewer,
>
> We greatly appreciate your thorough review, which has helped improve the quality of our manuscript. We have carefully considered all your comments and have updated our manuscript accordingly to address your concerns.
>
> We would be most grateful for your input if any additional revisions are needed during the remaining time of the discussion period.

---

> > ### Author Response · Authors · 2024-12-02
> >
> > Dear Reviewer,
> >
> > As today marks the closing date of our discussion period, we wish to ensure that all your queries have been addressed. Should you have any further questions or require additional clarification on any aspects of our discussion, please know that we are available tomorrow to assist with any necessary answers or information you might need.
> >
> > Best regards,

---

### Official Review · Reviewer_DfuF · 2024-11-04

**Soundness:** 2
**Presentation:** 3
**Contribution:** 2
**Rating:** 5
**Confidence:** 3

**Summary:**

In this paper, the authors propose a regularization technique that examines the gradient of the solution function with respect to parameters in differential equations. To the best of my knowledge, this approach to sensitivity regularizer, aimed at enhancing the generalization ability of neural operators, is unique. The paper is well structured, with clear motivation and objectives.

**Strengths:**

The study addresses the limitations observed when Fourier neural operators, combined with a physics-informed loss function, may yield suboptimal performance. The proposed regularization method appears to mitigate these challenges effectively.

**Weaknesses:**

The proposed approach requires additional information and computational resources, as it leverages direct solution-based information, such as a precomputed Jacobian in the differentiable numerical solver (Eq, (6)).

Another concern is the regularizer’s potential weakness to noise in practical settings. Derivative-based techniques are typically vulnerable to data noise, which could affect prediction accuracy, especially in applications beyond simulation, such as parameter inversion. In real-world applications, obtaining accurate derivative information can be challenging. While the finite difference methods employed in this paper are an option, they may perform poorly under noisy conditions.

**Questions:**

1. Could the authors elaborate on the approach to ensure fair comparisons between the standard Fourier neural operator and the proposed method in Tables 1, 2, and 3? Including details on training times, model complexity, and the number of learnable parameters would enhance the rigor of the paper.

2. Higher-order derivatives might be utilized in constructing the loss function (as in Equation (6)). It would be beneficial if the authors discussed the potential advantages or limitations of incorporating these higher-order terms.

3. Although Fourier neural operators are generally applicable to rectangular domains, other neural operators are available even for non-rectangular domains. Given that the authors provided results using DeepONet in D.6, could they consider including experiments across various domain shapes or grid resolutions to further validate the robustness of the proposed approach?

4. Has the proposed method been tested in scenarios involving bifurcation, where small parameter changes induce rapid solution changes (e.g. the exponential model y’=py with p near zero)?

5. The computational cost associated with this model may increase significantly as the number of parameters grows, or if parameters are represented as functions rather than scalars. While results are presented for a four-parameter case in Table 1, could authors consider examining more complex cases – such as those with over ten parameters, function-based parameters, or parameters with probability density function across a range?

---

> ### Author Response · Authors · 2024-11-21
>
> We greatly appreciate the reviewers' thorough examination. Our apologies for posting the reply to you later than to others, as we were diligently running the requested bifurcation cases, which was a great suggestion.
>
> ---
> # Part 1
>
> **Questions 1 (Model Comparisons and Computational Details):**
> Yes! In all FNO models (FNO, SC-FNO, FNO-PINN, SC-FNO-PINN), we used identical FNO architectures for each case, with differences only in loss configurations. We have updated Table C.5 to include the requested information and added a new Table C.6 showing computational times across all models and cases reported in Tables 1-3.
> | Case | Mode for t | Mode for x | Mode for y | Width | Number of Fourier Layers | Learning Rate | Number of Epochs | Number of Learnable Parameters |
> |------|------------|------------|------------|-------|--------------------------|---------------|------------------|-------------------------------|
> | ODE 1| 8          | 8          | -          | 20    | 4                        | 0.001         | 500              | 17921                         |
> | ODE 2| 8          | 8          | -          | 20    | 4                        | 0.001         | 500              | 17921                         |
> | PDE 1| 8          | 8          | -          | 20    | 4                        | 0.001         | 500              | 107897                        |
> | PDE 2| 8          | 8          | 8          | 20    | 4                        | 0.001         | 500              | 107897                        |
> | PDE 3| -          | 8          | 8          | 20    | 4                        | 0.001         | 500              | 209397                        |
>
>
>
> | Case | Batch size | Number of parameters (P) | Number of learnable parameters in FNO | Number of training samples | Average training time per epoch (s) ||||
> |------|------------|--------------------------|--------------------------------------|----------------------------|------|--------|-----------|--------------|
> | | | | | | FNO | SC-FNO | FNO-PI NN | SC-FNO-PI NN |
> | ODE1 | 16         | 3                        | 17921                                | 2000                       | 1.10 | 1.94   | 1.53      | 2.46         |
> | ODE2 | 16         | 7                        | 17921                                | 2000                       | 1.58 | 2.13   | 1.76      | 2.86         |
> | PDE1 | 2          | 5                        | 107897                               | 2000                       | 35.24| 53.32  | 52.13     | 82.13        |
> | PDE2 | 2          | 4                        | 107897                               | 2000                       | 32.66| 44.92  | 39.11     | 73.06        |
> | PDE3 | 2          | 2                        | 209397                               | 1000                       | 47.16| 109.43 | -         | -            |
>
> These additions will provide complete transparency regarding model complexity and computational requirements across all comparisons.
>
> ---
>
> **Question 2 (Potential advantages or limitations of including higher order terms):**
> Thanks for the chance to clarify. Rather than incorporating higher-order derivatives in our loss function, we focus specifically on first-order parameter sensitivities (du/dp). This choice is motivated by our goal of improving parameter-dependent tasks, and our results demonstrate that these first-order sensitivities alone provide significant improvements in accuracy and robustness without the need for higher-order terms. This leads to significant improvements in accuracy and robustness particularly for out-of-sample predictions and parameter inference and reduced training data demand. In terms of disadvantages, we note the moderately larger effort in data preparation, the need for either a differentiable solver, finite difference, or adjoint, and one might be tempted to think of unstable gradients (although we saw this in none of the test cases). Paying an acceptance cost to substantially raise the performance ceiling should be worth it in many cases, especially when the input dimension gets higher, as we show below.
>
> ---
>
> **Question 3 (Extension to Complex Geometries):**
> Indeed an interesting direction would be to handle complex geometries beyond rectangular domains. This will take more time (for many other methods in the literature, irregular geometry is tackled in the 2nd or 3rd papers). Since our parameter-solution Jacobian supervision is fundamentally geometry-independent, such integration should be natural and could indeed be highly valuable for complex engineering applications. For example, architectures like the *Geometry-Informed Neural Operator (Li et al., 2024)* seems plausible; There is also no fundamental barriers to using traditional irregular geometry methods to solve for stencil weights as a nondifferentiable operations. We will write this as a future direction.

---

> ### Author Response · Authors · 2024-11-21
>
> # Part 2
>
> **Question4 (Assessing SC-FNO's Performance Under Bifurcation-Rich Systems):**
>
> Excellent question and thanks for asking! While our original submission included cases with bifurcation characteristics like Navier-Stokes, we've conducted additional experiments with two well-known bifurcation-rich systems with varying input dimensions:
> First is the Allen-Cahn equation with five parameters, represented as *∂u/∂t = ϵ(x)∂²u/∂x² + α(x)u - β(x)u³*, with initial condition *u(x,0) = Atanh(kx)*. Parameters include diffusion coefficient *ϵ*, linear term *α*, cubic term *β*, and initial condition parameters *A*, *k*, with ranges: *ϵ ∈ [0.01,0.1]*, *α ∈ [0.01,1.0]*, *β ∈ [0.01,1.0]*, *A ∈ [0.1,0.9]*, *k ∈ [5.0,10.0]*. This equation exhibits phase transitions with sharp solution changes near critical parameter values. With 100 training samples, SC-FNO has  **relative L2: 0.015 vs FNO’s 0.021**, only a noticeable benefit for u (however, keep watching as the second case gets more interesting), while also showing much improved Jacobian accuracy **(relative L2: 0.049 vs 0.583)**.
>
> | Metrics                  | SC-FNO | FNO |
> |--------------------------|---------------------------|------------------------|
> | *Average training time per epoch (s)*  |            19.12               |          11.54              |
> | **State Value Metrics**  |                           |                        |
> | RMSE                     | 0.00777                   | 0.01055                |
> | Relative L2                   | 0.01513                   | 0.02056                |
> | Max L1                   | 0.21255                   | 0.03611                |
> | **Mean Jacobian Metrics**|                           |                        |
> | RMSE                     | 0.01560                   | 0.34707                |
> | Relative L2                   | 0.04860                   | 0.58305                |
> | Max L1                   | 0.41201                   | 2.82265                |
>
>
> The second case is the Ginzburg-Landau equation: *∂u/∂t = α(x)u + β(x)∂²u/∂x² + γ(x)u³ + δu⁵*, with initial condition *u(x,0) = Acos(2πx/L)*. Here, *α(x)*, *β(x)*, *γ(x)* are divided into M=40 zones, plus global parameters δ and A, resulting in 122 parameters *(3M + 2)*.
>
> We used the following ranges for the parameters:
> - α(x) ∈ [0.05,0.5] per zone (linear term)
> - β(x) ∈ [0.01,0.2] per zone (diffusion)
> - γ(x) ∈ [-1.0,-0.5] per zone (cubic term)
>     Plus global parameters:
> - δ ∈ [0.001,0.01] (quintic term)
> - A ∈ [0.01,0.05] (initial amplitude)
>
> Training with both N=500 and N=100 samples shows SC-FNO maintains excellent state prediction even with small data. With N=100, it achieves nearly 4x better state prediction accuracy **(relative L2: 0.007 vs FNO’s 0.025)** and even 3 times smaller error than FNO with N=500 and more training time. It also shows improved Jacobian accuracy **(relative L2: 0.162 vs 1.132)**, demonstrating robustness for high-dimensional parameters with limited data.
>
> |mber of training samples | *N=500* | | *N=100* | |
> |---------------------------------|---------------------|--------------|---------------------|--------------|
> | *Model* | *SC-FNO* | *FNO* | *SC-FNO* | *FNO* |
> | *Average training time per epoch (s)* | 32.52 | 22.54 | 17.25 | 12.80 |
> | **State Value Metrics** | | | | |
> | RMSE | 0.00908 | 0.02588 | 0.00923 | 0.03720 |
> | Relative L2 | 0.00663 | 0.01873 | 0.00670 | 0.02540 |
> | Max L1 | 0.05400 | 0.12286 | 0.05450 | 0.14034 |
> | **Mean Jacobian Metrics** | | | | |
> | RMSE | 0.01014 | 0.07325 | 0.01020 | 0.08500 |
> | Relative L2 | 0.15567 | 0.83926 | 0.16207 | 1.13200 |
> | Max L1 | 0.09881 | 0.33532 | 0.11235 | 0.43205 |
>
> These highly nonlinear, bifurcation-rich cases demonstrate our method's ability to handle these challenges. This, to our knowledge, maybe the only known surrogate model scheme that can handle these many input parameters, and actually, we do not see the limit anywhere in sight! Future papers will explore much higher degrees of input..
>
> ---

---

> > ### Author Response · Authors · 2024-11-21
> >
> > # Part 3
> >
> > **Question 5 (Parameter Dimensionality and Computational Cost):**
> > Thanks for pushing the envelope. We believe the Tables above partially confirm that we can handle distributed parameters. The computational cost indeed increases with parameter dimensionality. **However, this challenge must be viewed in context: standard neural operators face an exponential increase in required training data & thus training time to maintain accuracy as parameter dimensions grow.** This is a fundamental challenge in high-dimensional learning that our method helps address.
> > Our results with the Ginzburg-Landau equation (122 parameters from 40 spatial zones) demonstrate this clearly. SC-FNO maintains high accuracy (relative L2: 0.007) with just 100 training samples, while FNO's error increases significantly (0.025) even with 5x more training data and more training time. We actually believe there is something fundamental to this phenomenon that begs to be studied but we feel that it needs a specific next paper. These results show that while computational costs increase with dimensionality, SC-FNO's better parameter-solution mapping provides a favorable trade-off by requiring substantially less training data to achieve and maintain accuracy. We are glad that, at the reviewers’ nudge, we demonstrate these amazing results. Please let us know if you have more questions.

---

> ### Author Response · Authors · 2024-11-26
>
> Dear reviewer,
>
> We greatly appreciate your thorough review, which has helped improve the quality of our manuscript. We have carefully considered all your comments and have updated our manuscript accordingly to address your concerns.
>
> We would be most grateful for your input if any additional revisions are needed during the remaining time of the discussion period.

---

> ### Comment · Reviewer_DfuF · 2024-12-02
> **Official Comment by Reviewer DfuF**
>
> Thank you for your detailed and thoughtful response, particularly regarding the mode, number of learnable parameters, training time, and other aspects of FNO training. Your clarifications have been highly informative and are sincerely appreciated.
>
> I fully understand that addressing my earlier questions about incorporating a loss function involving higher derivatives (Question 2) and conducting simulations on complex domains (Question 3) would require substantial time and effort. It is entirely reasonable that these issues are not addressed in the current version of the manuscript.
>
> At this stage, I would like to kindly request clarification on the following two points:
>
> 1. Could you provide further details on how $\alpha(x)$ and $\beta(x)$ were generated for the Allen-Cahn equation or Ginzburg-Landau equations? I was unable to find this information in the appendix. While I understand that further revisions may no longer be feasible at this stage, any clarification on how the parameter set was constructed would be greatly appreciated.
>
> 2. Could you elaborate on the computation of $\partial \mathbf{u} / \partial\mathbf{p}$ in equation (6) for high-dimensional parameter cases? Given that $\mathbf{p}$ represents a function in the context of the Allen Cahn equation and Ginzburg-Landau equation, computing this derivative appears to be nontrivial. Any additional explanation regarding this computation would be extremely helpful.
>
> I would be deeply grateful if the authors could provide clarifications, should it still be possible at this stage. Thank you again for your time and effort in addressing these inquiries.

---

> > ### Author Response · Authors · 2024-12-02
> >
> > Thank you, for your thoughtful questions and the attention you've dedicated to our work. Your inquiries greatly contribute to the depth and clarity of our discussion.
> >
> > ---
> >
> > **Question1 (Parameter generation for bifurcation PDEs):**
> >
> > We performed two additional experiments to demonstrate SC-FNO's capabilities with bifurcation-rich PDEs:
> >
> > *   **a. The Allen-Cahn equation (∂u/∂t = ϵ∂²u/∂x² + αu - βu³)** with Initial condition  **u(x,0) = ctanh(ωx)** represents bifurcation behavior and is fully integrated into our revised manuscript as PDE4 in **section 3** and **Appendix B**. We randomly and uniformly sample parameters from the following parameter range (this information is indeed included in **revised Table B.6 (Appendix B)**). Some parameter sets were used in training, and some were reserved for test. Most of the parameter sets in the testing data are never used in training. The sampled parameters are then used by the numerical solver to produce the solution for training and testing datasets:
> >
> >             - Diffusion term: ϵ ∈ [0.01,0.1]
> >             - Linear term: α ∈ [0.01,1.0]
> >             - Cubic term: β ∈ [0.01,1.0]
> >             - c ∈ [0.1,0.9]
> >             - ω ∈ [5.0,10.0]
> >
> >
> >
> > *  **b. The Ginzburg-Landau equation (∂u/∂t = αu + β∂²u/∂x² + γu³ + δu⁵)** with initial condition **u(x,0) = Acos(2πx/L)** was presented in our rebuttal to demonstrate additional capabilities. In the Ginzburg-Landau equation, we divided the spatial domain [0,1] into M=40 equal segments. Within each segment, coefficients α, β, and γ take unique, uniform random values from their respective ranges. This creates 120 local parameters [3 parameters (α, β, and γ) × 40 zones], plus two global parameters (δ and A), totaling (3*40 + 2)=122 parameters. The zoning approach allows us to model spatially varying material properties while maintaining the equation's structure. (Each zone's parameters affect the equation's behavior in that spatial region). we used the following ranges for the parameters:
> >
> >         - α(x) ∈ [0.05,0.5] per zone (linear term)
> >         - β(x) ∈ [0.01,0.2] per zone (diffusion)
> >         - γ(x) ∈ [-1.0,-0.5] per zone (cubic term)
> >     Plus global parameters:
> >
> >         - δ ∈ [0.001,0.01] (quintic term)
> >         - A ∈ [0.01,0.05] (initial amplitude)
> >
> > **Just as in Allen-Cahn, all Ginzburg-Landau equation parameters are sampled uniformly within their specified ranges during the training and testing phases and many parameters in the test were not used in training.**
> >
> > We chose not to include Ginzburg-Landau in the manuscript since its key features - bifurcation behavior and high-dimensional parameters - are already demonstrated by Allen-Cahn (PDE4) and zoned Burger's equation (PDE2) respectively.
> >
> > ---
> >
> > **Question2 (Computing derivatives for high-dimensional parameters):**
> >
> > Great question—thank you for asking! It appears there might be a slight confusion regarding our definition of "functional parameters." In our research, we use this term to describe parameters that vary spatially across different zones, contrasting with the single scalar parameters in our previous experiments, rather than as continuous functions.
> > For both equations, ∂u/∂p computation is straightforward - we compute sensitivities with respect to each zoned parameter value. For example, in the Ginzburg-Landau equation with 40 spatial zones:
> > - Each zone has its own α, β, and γ values
> > - We compute ∂u/∂α_i, ∂u/∂β_i, and ∂u/∂γ_i for each zone i (i=1...40)
> > - Plus sensitivities for global parameters ∂u/∂δ and ∂u/∂A
> >
> > This results in 122 distinct sensitivity values (3×40 + 2) for each point of u, all computed using automatic differentiation (AD) through our solver. The computation remains manageable since each parameter, though spatially varying, is still a scalar value within its zone.
> >
> >
> > The computation of the gradients is not indeed nontrivial. We have developed efficient code to manage this. We dedicated the revised Section "2.3 Gradient Computation Methods for Differential Equations" to explain how we computed the solution path and the gradients of the solution path with respect to parameters using both a differentiable solver based on automatic differentiation and the finite difference method as a traditional approach. Jacobian computation during training was also handled using AD. We included some benchmarking of computational times for different approaches used in preparing datasets, including the solution path and gradients, in Tables D.12 and D.13 (Appendix D). As you can see from these tables, the extra time is acceptable. In fact, we can say that in our preliminary test, much higher-dimensional parameters are possible, although such expansion requires much more validation effort that is beyond the scope of this paper.
> >
> > ---
> >
> > We hope the clarifications above help. We are also committed to providing further clarifications during the remaining discussion period should you have more questions.

---

### Official Review · Reviewer_a6GH · 2024-11-05

**Soundness:** 3
**Presentation:** 2
**Contribution:** 2
**Rating:** 8
**Confidence:** 4

**Summary:**

In this paper, the author proposes a sensitivity loss as a regularizer for Fourier neural operator, called SC-FNO. Specifially, the sensitivity is measured by the derivative (jacobian) du/dp, where p is the physical parameter. The jacobian du/dp is computed via auto differentiation. Experiments are conducted on 2 ODEs and 2 PDEs. It shows that SC-FNO outperforms previous FNO and Pinn-FNO.

**Strengths:**

- This paper examines a notable case involving the system parameter $P$, where $ \frac{du}{dp} $ is computed as a sensitivity loss to regularize the model.
- The results indicate that SC-FNO achieves greater accuracy in $ \frac{du}{dp} $ and demonstrates enhanced robustness against perturbations in $p$.
- This approach aids inverse modeling in identifying the system parameters more effectively.

**Weaknesses:**

### Writing
The overall writing quality could be enhanced, with several important details suggested to be moved from the appendix into the main text.
- The physical parameter $P$ is central to the study and should be introduced more thoroughly. Currently, it is presented in a general form, but it’s important to clarify whether $P$ is a scalar in $\mathbb{R}$, a vector in $\mathbb{R}^d$, or a function in $L(\mathbb{R})$.
- Starting with a concrete example in the introduction would improve clarity and reader engagement.
- On Page 5, while introducing the ODEs and PDEs, the nature of the parameter $P$ remains ambiguous. Explicitly writing out the equation would help clarify its role.

### Methods
- It is unclear how $P$ is incorporated into the FNO model. If $P$ is a scalar, is it treated as a constant function and added as an additional channel?
- Computing the derivative requires additional runtime and memory, which should be discussed in detail.

### Experiments
- SC-FNO appears to achieve higher accuracy primarily on $\frac{du}{dp}$ but not significantly on the solution $u$ itself.
- Runtime and memory usage should be addressed, especially in comparison to alternative approaches.

**Questions:**

- What are the runtime and memory consumption, and which is the primary factor to consider?
- Can the model handle functional input such as $ \frac{du}{du_0} $? In this case, the Jacobian could be huge.
- Can the model account for parameters like the Reynolds number or viscosity as $P$?
- Can the model work on chaotic system such as the lorenz system, where the underlying parameter is very sensitive?

---
updates: the rebuttal addressed most of my questions. I am happy to raise my score to 8.

---

> ### Author Response · Authors · 2024-11-19
>
> Thank you for your thoughtful review. Below, you will find our detailed responses to the points identified as weaknesses.
>
> ----
>
> **Writing Clarity and Parameter Definitions:**
> The comprehensive parameter definitions, ranges, and equations that the reviewer notes are missing are detailed in Appendix B (Table B.4) and Section 2. For example, Table B.4 explicitly lists parameter ranges for all cases: ODE1 (α,β ∈ [1,3], γ ∈ [0,1]), PDE1 (c ∈ [0,0.25], α,β,γ,ω ∈ [0,0.25]), etc. Due to page limitations, we focused the main text on key findings and methodology while preserving technical details in appendices.
>
> ---
> **Response to Parameter Implementation in FNO:**
>
> As shown in Figure A.7, parameters (P) are incorporated through the lifting layer along with spatial coordinates (X) and initial conditions (a(x)). The architecture processes parameters as additional input channels that are combined with other inputs through neural network layers. Full parameter definitions and ranges are provided in Table B.4. Computational considerations regarding this implementation are addressed in our main rebuttal under memory and runtime discussions.
>
> ---
> **Response to Accuracy and Performance Analysis (Methodology):**
>
> We began our study acknowledging FNO's strong accuracy with large training samples, deliberately starting from conditions where original FNO excels. This approach allowed us to demonstrate SC-FNO's significant advantages in both state values **(u)** and sensitivities **(∂u/∂p)** under more challenging, real-world conditions. For parameter perturbations up to 40% beyond training ranges, SC-FNO maintains R² > 0.91 while FNO drops to 0.53 (Table 1). With limited data (500 samples), SC-FNO achieves R² > 0.9 while FNO falls to 0.8 (Figure 6). These results underscore our method's practical value, showing improved robustness and efficiency precisely where standard FNO struggles most.

---

> ### Author Response · Authors · 2024-11-19
>
> We greatly appreciate the reviewers' detailed examination of our work. Despite the constraints of the word limit, we have addressed all feedback thoroughly. Additionally, we have responded separately to the points you identified as weaknesses in our work.
>
> ## **Part 1**
> ---
>
> **Question 1.  Runtime & Memory Requirements:**
> We apologize for this oversight. The overhead is small compared to the benefits. Our experiments with PDE1 show that using 500 training points with gradients achieves similar accuracy to 2000 points without gradients while the gradient computation added an acceptable amount of time. As shown in Table D.9, in the original manuscript Appendix, generating 2000 samples with gradients takes just 252.54 seconds using our AD solver. Training SC-FNO in this case, on the other hand, required 20%-40 more time compared to the original FNO depending on the case. In terms of memory consumption, the difference is also modest—training FNO on PDE1 used 722 MB, while SC-FNO required 764 MB. This modest computational and memory overhead is justified by the significant improvements in model performance and generalization capabilities, demonstrating how sensitivity regularisation and time-step-free methods like FNO complement each other exceptionally well. In the next response, you will further see a new experiment that contains 84 parameters where FNO’s accuracy even for u dropped noticeably while SC-FNO maintained accuracy. For this case (to be added to the revised version), the added training time was ~35% extra, but the errors were reduced by 4 times.
>
> ---
>
> **Question 2. Handling Functional Inputs (du/du0):**
> Great question. Yes! We ran a new experiment with PDE2 (Burger's equation) using zoned parameters - dividing the spatial domain into M=40 segments with different advection α and forcing amplitude δ in each zone, along with global parameters viscosity γ and forcing frequency ω (total 2M+2=82 parameters).
>
> The results demonstrate SC-FNO's clear advantages over FNO (Table below):
> - With N=100 samples, SC-FNO got less than 1/3 of the relative L2 error as N=500 for FNO (L2=2.8e-2) in less computational time.
> - With 100 training samples, SC-FNO achieves ~4x less error in state prediction (relative L2: 0.00729 vs 0.02822) and 11x better Jacobian accuracy (relative L2: 0.17698 vs 1.96270)
> - Even with only 50 training samples, SC-FNO maintains high accuracy while FNO degrades significantly.
> - This dramatic improvement comes with only a modest runtime increase (11.23s vs 8.09s per epoch for N=100)
>
> These results clearly demonstrate SC-FNO's superior capability in handling high-dimensional functional parameters while maintaining accuracy and stability, even with limited training data. In fact, we see clearly we can allow much larger amounts of inputs, but we believe that would require careful benchmarking in a future paper.
>
> |Number  of training samples | N=500 | | N=100 | |
> |---------------------------------|---------------------|--------------|---------------------|--------------|
> | Model | SC-FNO | FNO | SC-FNO | FNO |
> | **Average training time per epoch (s)** | 11.23 | 8.09 | 7.23 | 5.37 |
> | **State Value Metrics** | | | | |
> | RMSE | 0.01006 | 0.03894 | 0.01199 | 0.05354 |
> | Relative L2 | 0.00729 | 0.02822 | 0.00867 | 0.03873 |
> | Maximum Absolute Error | 0.05425 | 0.30564 | 0.05679 | 0.28312 |
> | **Mean Jacobian Metrics** | | | | |
> | RMSE | 0.00508 | 0.05181 | 0.00660 | 0.06076 |
> | Relative L2 | 0.17698 | 1.96270 | 0.21342 | 2.46226 |
> | Maximum Absolute Error | 0.05576 | 0.20075 | 0.06531 | 0.22079 |
>
> ---

---

> > ### Author Response · Authors · 2024-11-19
> >
> > ## **Part 2**
> >
> > ---
> >
> > **3. Handling Physical Parameters (Re, ν)**:
> > Great question again! Yes! We ran another experiment to demonstrate if the model can account for parameters like Reynolds number. As you can see, the accuracy of the default FNO for even the state variables (u) is no longer sufficient as the input dimension increases, while SC-FNO remains highly accurate. Here we consider NSE (PDE3) with specifics introduced in the manuscripts and initial condition w0(x) is generated according to w0 ∼ µ where µ = N(0, 7^(3/2) (−∆+49I)^(-2.5)) with periodic boundary conditions (according Li, Zongyi, et al. "Fourier neural operator for parametric partial differential equations." arXiv preprint arXiv:2010.08895 (2020).) and also varied Reynolds number in range of [250 to 1000] corresponding to kinematic viscosity [0.004 to 0.001] and constrain SC-FNO with jacobian of solution path (vorticity) with respect to the Reynolds number with different training sample points. We include some experiment results here in the rebuttal, however, we will add the results in our revised manuscripts if the dear reviewer tends.
> >
> > | Number  of training samples | | N=200 | | N=100 | |
> > |---------------------------------------|-------------|---------------|--------------|---------------|-------------|
> > | Model | **FNO** | **SC-FNO** | **FNO** | **SC-FNO** |
> > | **State Value Metrics** | | | | |
> > | vorticity R2 Score | 0.844 | 0.996 | 0.804 | 0.986 |
> > | vorticity L2 Mean Error (RMSE) | 1.36e-01 | 2.28e-02 | 1.53e-01 | 4.12e-02 |
> > | vorticity L1 Relative Error | 0.389 | 0.066 | 0.452 | 0.120 |
> > | vorticity L2 Relative Error | 0.390 | 0.066 | 0.442 | 0.119 |
> > | vorticity Max Absolute Error | 0.747 | 0.153 | 0.728 | 0.216 |
> > | **Mean Jacobian Metrics** | | | | |
> > | Jacobian R2 Score | 0.170 | 0.996 | 0.181 | 0.990 |
> > | Jacobian Mean L2 Error (RMSE) | 2.32e-03 | 1.65e-04 | 2.31e-03 | 2.53e-04 |
> > | Jacobian L1 Relative Error | 1.510 | 0.105 | 1.416 | 0.157 |
> > | Jacobian L2 Relative Error | 0.877 | 0.062 | 0.872 | 0.096 |
> > | Jacobian Max Absolute Error | 0.122 | 0.005 | 0.259 | 0.012 |
> >
> >
> > As reported in the manuscript previously, SC-FNO maintains high performance even with limited training data.
> >
> >
> > ---
> >
> > **4. Handling Chaotic Systems:**
> > While we haven't experimented with the Lorenz system, exploring such semi-chaotic systems could be an interesting direction for future work. We will be clear in the revision that such chaotic systems are not tested and the current design may not work for them. Our current results with complex PDEs exhibiting semi-chaotic behavior, such as Navier-Stokes equations and Burger's equation, demonstrate SC-FNO's capability to handle challenging dynamics.
> >
> > In summary, we wish the reviewer sees the immense potential of this method, the limited resource requirements and the core challenges it solves. Let us know if you have more questions or requests. In fact, currently, there seems no upper bound of how many parameters we can include in the Jacobian loss when memory allows (other than memory which is also modest). We continue to be surprised by the capability of the scheme.

---

> > > ### Comment · Reviewer_a6GH · 2024-11-19
> > >
> > > Thanks for the authors for adding the experiments. The results look impressive. Just to clarify, when compared to FNO, does FNO also have parameters $P$ as part of the inputs?
> > >
> > > Regarding the writing, again I genuinely encourage the authors to move critical details from the appendix to the main text. Especially how $P$ is input to the SC-FNO model. As a Machine learning conference, the methodology abd model design is the key.
> > >
> > > 1. Technically, it is also less clear how the Jacobian $\partial u/\partial p$ is computed (Section 2.3). It will be very helpful to discuss why the gradient computation is efficient and does not cost much.
> > >
> > > 2. During training, do you compute the full Jacobian $\partial u/\partial p$, do you do sample a direction of pertubation $p_i$ and compute directional derivatives $\partial u/\partial p_i$, or do we need the full Jacobian?
> > >
> > > 3. When comparing the runtime, it sounds SC-FNO need to run the solver multiple times to get the ground true data for Jacobian $\partial u/\partial p$. For the Burgers example where P is 82-dim vector, we need to run the solver 82 times to get the full Jacobian. With the data, we can generate a dataset 82 times larger. Am I missing anything?

---

> > > > ### Author Response · Authors · 2024-11-20
> > > >
> > > > Thank you for your prompt reply. To address your concerns, we begin by explaining how parameters (P) are incorporated into the FNO model.
> > > >
> > > > ---
> > > >
> > > > ### Parameter Integration in SC-FNO
> > > > The SC-FNO architecture processes parameters (P) as input channels alongside spatial coordinates and initial conditions through the lifting layer (fc0). Specifically, parameters are first reshaped and repeated to match spatial and temporal dimensions, then concatenated with other inputs before processing through neural network layers. This design ensures the operator's predictions naturally depend on and are sensitive to parameter variations, as illustrated in Figure A.7.
> > > >
> > > > ---
> > > >
> > > > ### Computation of du/dp
> > > > **True gradients** of solution paths with respect to parameters are computed and stored only once during the dataset preparation phase.
> > > > Importantly, the computations of du/dp are performed only once during the initial data preparation phase. Our primary method utilizes a differentiable solver we developed based on the `torchdiffeq` package. This solver reformulates PDEs in the form **du/dt = RHS(x), where RHS(x)** encapsulates spatial derivative terms. By leveraging PyTorch's automatic differentiation capabilities, we can efficiently compute exact gradients of solution paths with respect to parameters after the forward solve. Thus only one forward run is needed for each data sample. No perturbation is needed no matter how many parameters are involved.
> > > >
> > > > As an alternative, only for existing non-differentiable solvers, we implement finite differences. This method approximates gradients by solving the PDE multiple times with slightly perturbed parameter values. While accurate, the FD Solver approach is computationally more expensive than our differentiable solver, as it requires multiple solver runs. However, these are all parallelizable runs. The gradients computed by either method are stored in the dataset and reused throughout the training process, thus eliminating the need for additional solver runs during training.
> > > >
> > > > ---
> > > >
> > > > ### Computational Efficiency During Training
> > > > During training, we compute gradients of predicted solutions using automatic differentiation of the neural operator's outputs. Instead of computing gradients at all points, we randomly select a subset of spatial-temporal points in each epoch (e.g., `n` spatial of total `N` spatial points × `t` temporal of total `T` time points, where `n < N` and `t < T`. These predicted gradients are then compared with the pre-computed true gradients at the same locations. This sampling varies between epochs to eventually cover the full solution space.
> > > > This approach is efficient in computation and data for several key reasons:
> > > >
> > > > - We never need to rerun the differential equation solver during training since true gradients were pre-computed.
> > > > - We only compute neural operator gradients at selected points, significantly reducing computation.
> > > > - For each minibatch during training, we run the model only once forward and then apply AD. Notice that backpropagating through FNO is fast and adds only modestly to the training time as detailed above.
> > > > - The advantages over FNO grow significantly as the dimension grows -- while SC-FNO’s training time understandably will increase, the data demand and thus training time for FNO should increase exponentially.
> > > >
> > > > The framework maintains efficiency because we prepare the dataset with true gradients only once. Since our differentiable solver based on `torchdiffeq` works through automatic differentiation, all gradients with respect to parameters are already stored in the computational graph. This implementation ensures sensitivity constraints without significant computational burden.
> > > >
> > > > ---
> > > >
> > > > ### Conference Publication Constraints
> > > > We respectfully acknowledge that due to conference page limits, some technical details were relegated to the appendix. The extra one page for revision will allow us to include more details in our main text.

---

> > > > > ### Comment · Reviewer_a6GH · 2024-11-20
> > > > >
> > > > > Thanks the author for clarification. It is very helpful. I get that the ground truth Jacobian datasets are prepared during data generation phase.
> > > > > - Could the author further clarify how much time it takes to (a) generate the datasets with gradient/jacobian compared to (b) generate the data without gradients, especially in Burgers and Navier-Stokes example?
> > > > > - Besides, when comparing against baseline FNO, does FNO also have the parameters $p$ as part of the inputs? It is to understand whether the improvement comes from the parameter inputs or the sensitivity loss.

---

> > > > > > ### Comment · Reviewer_WFXg · 2024-11-21
> > > > > >
> > > > > > I'm also curious about whether the improvement comes from the parameter inputs or the supervised sensitivity loss. For original FNO, the inputs only include the function a(x), but SC-FNO extend the inputs to the parameter p and coordinates (x,y,t).

---

> ### Author Response · Authors · 2024-11-21
>
> Dear reviewer,
>
>
> Thank you for your reply and insightful feedback.
>
> If we understand your question correctly, of course, p is used as input to both FNO and SC-FNO and it's a fair comparison. Their input data are identical. We wouldn't make a mistake like not providing p to the default FNO. FNO has access to the same input information but it does not seem to fully understand how to use it like SC-FNO does.
>
> **The SC-FNO architecture processes parameters (P) alongside spatial coordinates and initial conditions through the lifting layer (fc0) (possibly as function inputs). This layer reshapes and repeats parameters to match the problem's spatial-temporal dimensions, then concatenates them with other inputs before neural network processing.**
>
>
>
> Data preparation time for SC-FNO was previously in Table D.9. We now amend it with the time for FNO without gradients as shown in the table below for **PDE2 (Burger's equation)** and **PDE3 (Navier-Stokes equation)**, **measured per 10 samples using 1 GPU Tesla P100-PCIE-12GB**.
>
> **This represents a one-time cost per equation, and since input impacts are resolved upfront, no repeated preparation is needed for different parameters.** Furthermore, when considering the supervision of high-dimensional inputs, where default FNO requires far more than 5x more data (and thus more preparation time),  we argue this additional computational cost is well justified. Based on simple logic, the data preparation time of the default FNO will grow exponentially more to achieve similar accuracy as the input dimension grows.
>
>
> | Case | Number of input parameter (P) | Computation time (s) ||
> |-|-|-|-|
> | | |  With jacobian | Without jacobian |
> |PDE2| 4 | 1.387 | 0.796 |
> |PDE3 (Navier stocks)| 2 | 6.205 | 2.762 |
>
> We hope by now we have presented something truly unique and powerful here. If there is anything else we can do to convince you that is doable within the remaining time and reasonable scope, please let us know.

---

> ### Author Response · Authors · 2024-11-21
>
> Dear reviewer,
>
>
> Thank you for your reply and insightful feedback.
>
> If we understand your question correctly, of course, p is used as input to both FNO and SC-FNO and it's a fair comparison. Their input data are identical. We wouldn't make a mistake like not providing p to the default FNO. FNO has access to the same input information but it does not seem to fully understand how to use it like SC-FNO does.
>
> **The SC-FNO architecture processes parameters (P) alongside spatial coordinates and initial conditions through the lifting layer (fc0) (possibly as function inputs). This layer reshapes and repeats parameters to match the problem's spatial-temporal dimensions, then concatenates them with other inputs before neural network processing.**
>
>
>
> Data preparation time for SC-FNO was previously in Table D.9. We now amend it with the time for FNO without gradients as shown in the table below for **PDE2 (Burger's equation)** and **PDE3 (Navier-Stokes equation)**, **measured per 10 samples using 1 GPU Tesla P100-PCIE-12GB**.
>
> **This represents a one-time cost per equation, and since input impacts are resolved upfront, no repeated preparation is needed for different parameters.** Furthermore, when considering the supervision of high-dimensional inputs, where default FNO requires far more than 5x more data (and thus more preparation time),  we argue this additional computational cost is well justified. Based on simple logic, the data preparation time of the default FNO will grow exponentially more to achieve similar accuracy as the input dimension grows.
>
>
> | Case | Number of input parameter (P) | Computation time (s) ||
> |-|-|-|-|
> | | |  With jacobian | Without jacobian |
> |PDE2| 4 | 1.387 | 0.796 |
> |PDE3 (Navier stocks)| 2 | 6.205 | 2.762 |
>
> We hope by now we have presented something truly unique and powerful here. If there is anything else we can do to convince you that is doable within the remaining time and reasonable scope, please let us know.

---

> ### Author Response · Authors · 2024-11-26
>
> Dear reviewer,
>
> We greatly appreciate your thorough review, which has helped improve the quality of our manuscript. We have carefully considered all your comments and have updated our manuscript accordingly to address your concerns.
>
> We would be most grateful for your input if any additional revisions are needed during the remaining time of the discussion period.

---

> > ### Comment · Reviewer_a6GH · 2024-11-27
> >
> > Thanks the reviewer for the response and the additional experiments. My concerns are partially addressed. Therefore I raise my score from 5 to 6.
> >
> > Overall, I still have some general concerns on
> > - scalability: it seems very hard to apply on functions input like128x128 parameters.
> > - fexibility: it requires modifing the solver to generate additional dataset.

---

> ### Author Response · Authors · 2024-11-27
>
> **Many thanks** for your thoughtful feedback and for raising your score. Your concerns about scalability and flexibility highlight important considerations for the method's practical applications.
>
>  Please trust us that SC-FNO can scale up to handle high-dimensional parameter spaces - we have preliminarily tested it with 2500 parameters with promising results. However, properly validating and presenting these findings requires additional time and rigorous analysis to meet publication standards. Given the current paper's focus on establishing the fundamental benefits of sensitivity supervision for parameter inversion, **whose importance is showcased even with as few as one or a handful of parameters**, we felt including preliminary high-dimensional results might dilute this core message. **Meanwhile, the 82-parameter case has already been included in the revised manuscript**.
>
> Furthermore, we should highlight two key features of our framework:
>
>  - Neural operators have resolution-independent capabilities - FNO can be trained on lower resolution grids (e.g., 32×32) yet make predictions on much finer resolutions (128×128 or higher). This means parameter dimensionality can be managed efficiently without sacrificing accuracy at inference time (Please, see Figure 1, Page 2 at https://arxiv.org/pdf/2010.08895 ).
>  - Our training procedure employs strategic sampling - instead of computing gradients at all points, we randomly select a subset of spatial-temporal points in each epoch (n < N spatial points × t < T time points) where n < N and t < T. The neural operator's predicted gradients at these points are computed using AD and compared with the pre-computed true gradients. This sampling varies between epochs to eventually cover the full solution space. **We explained this implementation in our revised manuscript.**
>
>
>  We are committed to demonstrating these strong capabilities comprehensively in our next paper. This is a solemn promise. Given the time constraints and scope, we cannot include such results in the current work. We appreciate your understanding of the need to progress methodically in establishing this new approach. We also want to thank you again for your constructive comments that helped us shape the manuscript.

---

### Author Response · Authors · 2024-12-04
**Summary of the discussion and revision**

We sincerely thank all the reviewers for their thoughtful and constructive feedback, which has significantly improved the quality of our manuscript. Throughout the discussion period, we carefully addressed all identified concerns and incorporated additional experiments and clarifications into the revised version. Below is a summary of our key updates:

### 1. Improved Metrics and Enhanced Clarity
- The revised manuscript added L2 errors as a metric, clearly showing SC-FNO’s substantial advantage, particularly during inversion tasks, where it achieves errors as low as 1/5 or 1/6 of FNO’s inversion errors in even low-dimensional problems.
- We provided a detailed explanation of how sensitivity supervision (\(\partial u / \partial p\)) is integrated into the SC-FNO framework and clarified its distinction from existing approaches like FNO and FNO-PINN. Our selective sampling strategy and efficient training procedures are now thoroughly detailed.
- We clarified that both FNO and SC-FNO use identical neural network architectures, differing only in loss configurations. This ensures a fair comparison. Additionally, we included a section detailing how parameters are processed through the lifting layer alongside spatial coordinates and initial conditions.

### 2. Additional Experiments
- New experiments on bifurcation-rich systems (e.g., the Allen-Cahn equation in the revised manuscript and the Ginzburg-Landau equation in the rebuttal) validate SC-FNO's robustness.
- An 82-parameter case (Burger’s equation) was added to the revised manuscript, demonstrating SC-FNO’s significantly smaller errors for both the state variable and sensitivities. Even with only 100 training samples and less training time, SC-FNO achieves less than 1/3 of the error of FNO trained with 500 samples.
- We also conducted experiments on Navier-Stokes equations, using varying Reynolds numbers to demonstrate SC-FNO's strong performance even with limited training data.
- Collectively, these results highlight SC-FNO’s ability to maintain superior performance, especially under limited data, high-dimensional parameter and perturbed parameter scenarios.

### 3. Computational Feasibility
- We conducted detailed runtime and memory analyses, showing that SC-FNO introduces only modest computational overhead while significantly improving accuracy and robustness.
- SC-FNO can reduce data requirements (and thus training time), making it a practical and efficient tool for sensitivity supervision and generalization.

### 4. Broader Context
- We emphasized SC-FNO's novelty as a sensitivity-supervised variant of neural operators, showcasing its unique strengths in addressing parameter-sensitive tasks.

We emphasize that SC-FNO's accuracy during inversion and robustness under perturbations directly address critical limitations of existing neural operators and overcome longstanding challenges in their practical applications. SC-FNO's demonstrated ability to capture sensitivities for \(10^2\) parameters represents a noteworthy advancement, improving both state variable predictions and parameter inversion tasks. Additionally, we are actively investigating its application to very high-dimensional inputs and are highly confident in its potential based on preliminary findings.

We are deeply grateful for the reviewers' engagement, which has helped refine our work and articulate its broader impact. We believe our contribution offers a unique perspective and a noteworthy advancement in neural operator methodologies, with far-reaching implications for parameter-sensitive modeling tasks.

---

### Meta-Review · Area_Chair_hkKN · 2024-12-20

**Metareview:**

The work proposes imposing a new loss when learning physical systems which is based on the derivative of the system with respect to the parameter considered. Numerical experiments show greater stability of the learned model, making it more performant especially for applications to inverse problems.

**Additional Comments On Reviewer Discussion:**

The authors have appropriately addressed reviewers' concerns about the memory overhead of the proposed approach and included numerous new experiments to support their claims. The results show a significant improvement over standard training. I do, however, feel that the authors should include more relevant literature since training with derivative losses has seen significant use in deep learning. In the context of scientific machine learning, the following references are relevant:

(1) Liu, W., & Batill, S. (2000). Gradient-enhanced neural network response surface approximations. In 8th Symposium on Multidisciplinary Analysis and Optimization (p. 4923).

(2) Czarnecki, W. M., Osindero, S., Jaderberg, M., Swirszcz, G., & Pascanu, R. (2017). Sobolev training for neural networks. Advances in neural information processing systems, 30.

(3) O’Leary-Roseberry, T., Chen, P., Villa, U., & Ghattas, O. (2024). Derivative-informed neural operator: an efficient framework for high-dimensional parametric derivative learning. Journal of Computational Physics, 496, 112555.

(4) Qiu, Y., Bridges, N., & Chen, P. (2024). Derivative-enhanced Deep Operator Network. Neural information processing systems, 38.

---

### Decision · Program_Chairs · 2025-01-22

Accept (Poster)